

# Beyond $\mathcal{PT}$-symmetry: Towards a symmetry-metric relation for time-dependent non-Hermitian Hamiltonians

**Luís F. Alves da Silva, Rodrigo A. Dourado and Miled H. Y. Moussa**

Instituto de Física de São Carlos, Universidade de São Paulo,
P.O. Box 369, São Carlos, 13560-970, SP, Brazil

## Abstract

In this work we first propose a method for the derivation of a general continuous antilinear time-dependent (TD) symmetry operator $I(t)$ for a TD non-Hermitian Hamiltonian $H(t)$. Assuming $H(t)$ to be simultaneously $\rho(t)$-pseudo-Hermitian and $\Xi(t)$-anti-pseudo-Hermitian, we also derive the antilinear symmetry $I(t) = \Xi^{-1}(t)\rho(t)$, which retrieves an important result obtained by Mostafazadeh [J. Math, Phys. 43, 3944 (2002)] for the time-independent (TI) scenario. We apply our method for the derivation of the symmetries associated with TD non-Hermitian linear and quadratic Hamiltonians. The computed TD symmetry operators for both cases are then particularized for their equivalent TI Hamiltonians and $\mathcal{PT}$-symmetric restrictions. In the TI scenario we retrieve the well-known Bender-Berry-Mandilara result for the symmetry operator: $I^{2k} = 1$ with $k$ odd [J. Phys. A 35, L467 (2002)]. The results here derived allow us to propose a useful symmetry-metric relation for TD non-Hermitian Hamiltonians. From this relation the TD metric is automatically derived from the TD symmetry of the problem. Then, when placed in perspective with the antilinear symmetry $I(t) = \Xi^{-1}(t)\rho(t)$, the symmetry-metric relation finally allow us to derive the $\Xi(t)$-anti-pseudo-Hermitian operator. Our results reinforce the prospects of going beyond $\mathcal{PT}$-symmetric quantum mechanics making the field of pseudo-Hermiticity even more comprehensive and promising.

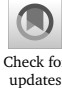
doi:10.21468/SciPostPhysCore.5.1.012

# 1   Introduction

In the last two decades, since the seminal contributions of Bender and Boettcher [1] and Mostafazadeh [2–4], $\mathcal{PT}$-symmetric Hamiltonians —invariant under parity ($\mathcal{P}$) and time-reversal ($\mathcal{T}$) symmetry— have been investigated in practically all domains of physics, from low to high energies, revealing to be an increasingly autonomous and thought-provoking field. The $\mathcal{PT}$-symmetry condition, weaker than Hermiticity, greatly expands the possibility of the Hamiltonian description of physical systems (with real eigenvalues [1] and conservation of the norm [2]), which is one of the strong calls for the field. And much has been done recently, such as the experimental realizations of Floquet $\mathcal{PT}$-symmetric systems [5] and $\mathcal{PT}$-symmetric flat bands [6], besides enhanced sensing based on $\mathcal{PT}$-symmetric electronic circuits [7] and $\mathcal{PT}$-symmetric topological edge-gain effect [8]. The linear response theory for a pseudo-Hermitian system-reservoir interaction was developed [9], as well as a protocol to approach non-Hermitian non-commutative quantum mechanics [10].

    In this work we propose a method to derive a general time-dependent (TD) continuous symmetry operator for a TD non-Hermitian Hamiltonian. This will be done in the broader scenario of non-autonomous Hamiltonians, and for this reason we revisit the TD non-Hermitian Hamiltonians of a cavity field under linear [11] and parametric [12] amplifications. These Hamiltonians have been considered for approaching TD non-Hermitian Hamiltonians under TD Dyson maps, thus extending the method proposed by Mostafazadeh [2]. This extension was also undertaken in Refs. [13,14]. The consequences of allowing the Hilbert space to have a TD metric were posed in Ref. [15], while a treatment of a TD non-Hermitian Hamiltonian

through a TI metric operator was done in Ref. [16]. The investigation of TD metric operators for the treatment of TD non-Hermitian Hamiltonians was undertaken in Ref. [17].

Many interesting contributions to the subject of TD non-Hermitian Hamiltonians have been presented [18–23]. We mention, in particular, a method that adds to the achievements of Ref. [11], enabling the unitarity of the time-evolution and the observability of non-Hermitian Hamiltonians through particular TD Dyson maps that define time-independent (TI) metric operators [24]. Moreover, we stress the introduction of the all-creation and all-annihilation TD pseudo-Hermitian bosonic Hamiltonians [25,26], able to generate an infinite squeezing degree at a finite time. A TD pseudo-Hermitian Hamiltonian for a cavity mode with complex frequency is also able to generate an infinite squeezing at a finite time [27]. Finally, we mention the enhancement of photon creation through the pseudo-Hermitian dynamical Casimir effect [28].

Our subject, pseudo-Hermiticity beyond $\mathcal{PT}$-symmetry, is in fact at the foundations of pseudo-Hermitian quantum mechanics. A theorem by Mostafazadeh [4] —formulated for TI non-Hermitian Hamiltonians, symmetries and metric operators— asserts that a *diagonalizable (non-Hermitian) Hamiltonian is pseudo-Hermitian if an only if it has an antilinear symmetry, i.e., a symmetry generated by an invertible antilinear operator*. Moreover, Bender, Berry and Mandilara [29] have shown that a non-Hermitian Hamiltonian presents a real spectrum not only when invariant under $\mathcal{PT}$-symmetry, but also under any antiunitary operator $I$ satisfying $I^{2k} = 1$ with $k$ odd. We also mention the demonstration that supersymmetry gives rise to non-$\mathcal{PT}$-symmetric families of complex potentials with entirely real spectra [30], and also the proposition of chiral metamaterials with $\mathcal{PT}$ symmetry and beyond [31]. Despite the generality of the pseudo-Hermitian requirement, the particular case of $\mathcal{PT}$-symmetric Hamiltonians gained prominence due to Bender and Boettcher's seminal work and certainly due to the strong physical appeal of parity and time-reversal invariance.

Our objective is precisely to explore more general symmetries than $\mathcal{PT}$ starting from the general scenario of TD non-Hermitian Hamiltonians. The method we propose for the derivation of TD symmetries for TD non-Hermitian Hamiltonians applies indistinctly to *linear* or *antilinear*, *unitary* or *nonunitary symmetries*. However, we assume the symmetry to be an *antilinear* operator aiming to retrieve the results by Mostafazadeh [4] and Bender-Berry-Mandilara [29] in the particular case of a TI scenario, i.e., TI non-Hermitian Hamiltonians, metrics and symmetries. The above mentioned theorem by Mostafazadeh [4] is retrieved when considering an *antilinear symmetry* while the result by Bender-Berry-Mandilara is retrieved when considering a *unitary antilinear* or *antiunitary symmetry*.

After presenting our method to derive the symmetry operator, we then apply it for TD non-Hermitian linear and quadratic Hamiltonian, modelling a cavity field under linear and parametric amplifications. As expected, we have derived TD continuous antilinear symmetries far more complex than the spatial reflection and time reversal. These TD symmetries are then particularized to the equivalent TI non-Hermitian linear and quadratic Hamiltonians and their $\mathcal{PT}$-symmetric restrictions. Then, in the TI scenario, the results in Refs. [2, 3, 29] are perfectly retrieved. The $\mathcal{PT}$-symmetry is then a particular case of more general symmetries in which spatial reflection is generalized to continuous rotations followed by additional displacement and/or squeezing in phase space. Our results reinforce the prospects of going beyond $\mathcal{PT}$-symmetric quantum mechanics making the field of pseudo-Hermitian quantum mechanics even more comprehensive and promising.

In addition, guided by the results in Refs. [2, 3, 29], here we explore the connection between *antilinear symmetries* and *metrics*. We derive a relation between the TD symmetry and a pair of TD metric operators, one linear and the other antilinear, which is analogous to the Mostafazadeh's relation [4] for the TI scenario. This connection between symmetry and metric is explored a little further, leading us to propose a relation between symmetry ($I$) and metric ($\rho$) operators. Then, this symmetry-metric relation is put in perspective with the TD

antilinear symmetry $I(t) = \Xi^{-1}(t)\rho(t)$ we have derived, allowing us to finally compute the $\Xi(t)$-anti-pseudo-Hermitian operator.

We stress that from the 1990s onwards, the field of radiation-matter interaction underwent extraordinary progress when experimentalists began to coherently control the process of the interaction of a single atom with a single photon of the radiation field [32,33]. This control allowed probing fundamental aspects of quantum mechanics and implementing quantum logic operations, among other important achievements. We also remember the construction of the Bose-Einstein condensates, which allowed unprecedented control in the manipulation of many-body processes [34,35]. For some time now, this control of the atom-field interaction has been sought towards time-dependent processes of radiation-matter interaction, as we know from the many advances made in the experimental verification of the dynamical Casimir effect [36–40]. We have reason to believe that not only the dynamical Casimir effect, but other processes involving TD Hamiltonians —such as a TD Josephson-type coupling in two-mode Bose-Einstein condensates [41,42]— will soon be achieved. Therefore, the TD pseudo-Hermitian quantum mechanics, in its most general form, accounting for TD non-Hermitian Hamiltonians, symmetries and metric operators, must be studied to account for these TD processes.

We also note that our results shed light on the treatments already presented in the literature on TD non-Hermitian Hamiltonians. For example, our developments considerably broaden our understanding of those presented in Refs. [11,12,25–28], where all the analysis on symmetry is reduced to the conditions for a TD Hamiltonian to be $\mathcal{PT}$-symmetric. From our conclusions, we now know that there are close connections between the group algebra of the Hamiltonian, the symmetry and the metric operator. As concluded below, when considering a TD non-Hermitian and non-$\mathcal{PT}$-symmetric Hamiltonian, we can now start by computing the symmetry operator $I(t)$ of the system modeled by the Hamiltonian $H(t)$, from which we automatically compute the metric operator, the $\rho$-pseudo-Hermitian and the $\Xi$-anti-pseudo-Hermitian operators.

Our paper is organized as follows. In Section 2 we briefly revisit the foundations of pseudo-Hermitian quantum mechanics for TI and TD Hamiltonians. In Section 3 we present a method for the construction of a general TD symmetry operator for a TD non-Hermitian Hamiltonian. In Section 4 we assume that the TD non-Hermitian Hamiltonian is simultaneously $\rho(t)$-pseudo-Hermitian and $\Xi(t)$-anti-pseudo-Hermitian, to derive the relation $I(t) = \Xi^{-1}(t)\rho(t)$ for our TD antilinear symmetry operator. From this relation we retrieve the Mostafazadeh's theorems for the TI scenario [4]. The TD non-Hermitian Hamiltonian describing a cavity field under linear amplification is introduced in Section 5. We then compute the TD symmetry operator for this non-Hermitian linear Hamiltonian using the method presented in Section 3. We demonstrate that this TD symmetry reduces to the $\mathcal{PT}$ operator when the non-Hermitian linear Hamiltonian is assumed to be $\mathcal{PT}$ symmetric. An ansatz for the Dyson map is then proposed for the construction of the pseudo-Hermiticity relation. In Section 6 we address the TI equivalent of the TD non-Hermitian Hamiltonian introduced in Section 5. In this TI scenario we retrieve the Bender-Berry-Mandilara [29] result, and when considering a $\mathcal{PT}$-symmetric TI non-Hermitian Hamiltonian, we verify that the TI symmetry again reduces to the $\mathcal{PT}$ operator. In Section 7 we introduce the TD pseudo-Hermitian Hamiltonian of a cavity field under parametric amplification. The TD symmetry operator is then derived. Also in this section the TD Dyson map is presented and the Hermitian counterpart of the Hamiltonian is computed. In Section 8 we consider the TI equivalent of the Hamiltonian introduced in Section 7. In Section 9 we present a symmetry-metric relation for TD non-Hermitian Hamiltonians by which the TD metric is automatically computed from the TD symmetry of the problem. By contrasting the symmetry-metric relation with the antilinear symmetry $I(t) = \Xi^{-1}(t)\rho(t)$ derived in Section 4, we are finally able to compute the $\Xi(t)$-anti-pseudo-Hermitian operator. In Section 10 we present our conclusions.

## 2 Pseudo-Hermiticity for TD and TI non-Hermitian Hamiltonians

We start our review following the Ref. [11], where a method is presented for approaching the quantum mechanics of TD non-Hermitian and non-observable Hamiltonians with TD metric operators. Alternative developments for the TD scenario are also given in Refs. [13,14]. From the particularization of these results for TI non-Hermitian Hamiltonians and metric operators, we then derive the results presented by Mostafazadeh in Ref. [2]. Considering a TD non-Hermitian Hamiltonian $H(t)$ and a nonunitary TD Dyson map $\eta(t)$, the Schrödinger equation $i\partial_t |\Psi(t)\rangle = H(t) |\Psi(t)\rangle$ ($\hbar = 1$) is transformed to $i\partial_t |\psi(t)\rangle = h(t) |\psi(t)\rangle$, with

$$h(t) = \eta(t)H(t)\eta^{-1}(t) + i\left(\frac{\partial}{\partial t}\eta(t)\right)\eta^{-1}(t), \tag{1}$$

and $|\psi(t)\rangle = \eta(t)|\Psi(t)\rangle$. This transformed Hamiltonian $h(t)$ becomes Hermitian as long as the TD pseudo-Hermiticity relation

$$H^\dagger(t)\rho(t) - \rho(t)H(t) = i\partial_t\rho(t) \tag{2}$$

is satisfied, where $\rho(t) = \eta^\dagger(t)\eta(t)$ is the TD metric operator ensuring the norm-conservation:

$$\left\langle \Psi(t) \big| \tilde{\Psi}(t) \right\rangle_{\rho(t)} = \left\langle \Psi(t) | \rho(t)\, \tilde{\Psi}(t) \right\rangle = \left\langle \psi(t) \big| \tilde{\psi}(t) \right\rangle. \tag{3}$$

In the same way that Eq. (2) ensures the norm-conservation —through the time derivative of Eq. (3)—, it also ensures the similarity transformation

$$O(t) = \eta^{-1}(t)o(t)\eta(t), \tag{4}$$

between the observables $O(t)$ and $o(t)$ in the pseudo-Hermitian and Hermitian systems, respectively, thus enabling the computation of the matrix elements

$$\left\langle \Psi(t) |O(t)| \tilde{\Psi}(t) \right\rangle_{\rho(t)} = \left\langle \Psi(t) |\rho(t)O(t)| \tilde{\Psi}(t) \right\rangle = \left\langle \psi(t) |o(t)| \tilde{\psi}(t) \right\rangle. \tag{5}$$

The reason why a TD Dyson map is required for the construction of the Hermitian counterpart $h(t)$ of an equally TD non-Hermitian $H(t)$, is to avoid unwanted constraints between the parameters defining $H(t)$. When considering a TI non-Hermitian $H$ so that an equally TI Dyson map $\eta$ can be considered, as in Ref. [2], the TD Dyson relation (1) simplifies to the similarity transformation

$$h = \eta H \eta^{-1}, \tag{6}$$

whereas the Eq. (2) simplifies to the well-known pseudo-Hermiticity relation

$$H^\dagger\rho = \rho H. \tag{7}$$

We now analyze the consequences of the TD extension of the Mostafazadeh's method based on the last two equations (6) and (7). First, within the TD extension and then the TD Dyson relation (1), we lose the similarity transformation (6) which ensures the observability of the Hamiltonian. However, the similarity transformation (4) remains valid for all operators $O(t)$ other than the Hamiltonian, making the TD pseudo-Hermitian Hamiltonians as pertinent as their TI partners. It is worth noting that the observability of TD Hamiltonians is a sensitive point even in Hermitian quantum mechanics, where, as discussed in Ref. [24], the Hamiltonian acts essentially as the generator of the model's dynamics.

We end this brief review noting that in Ref. [24] a method is proposed for the derivation of particular TD Dyson maps which ensures the observability of TD pseudo-Hermitian Hamiltonians —as much as in Hermitian quantum mechanics— by restoring the similarity transformation between $H(t)$ and $h(t)$. In the treatment we have developed below, however, we are not considering the method in [24], and the Hamiltonians $H(t)$ and $h(t)$ must be transformed through the TD pseudo-Hermiticity relation (2).

# 3  A Method for the construction of a general TD symmetry operator

In order to explore more general symmetries than $\mathcal{PT}$ for a TD non-Hermitian Hamiltonian $H(t)$, we first propose a method to derive this symmetry $I(t)$ which applies indistinctly to linear or antilinear, unitary or nonunitary symmetries. However, as anticipated above, from now on we assume this symmetry to be antilinear so that we can retrieve the results in Refs. [4,29] for the particular case of TI Hamiltonians and symmetries. Moreover, as well as the Hamiltonian, we assume the symmetry to be a TD operator. Starting from the Schrödinger equation for $H(t)$, we apply the antilinear operator $I(t)$ on both its left-hand sides and then replace $t$ by $-t$, to obtain

$$i\frac{\partial}{\partial t}I(-t)|\psi(-t)\rangle = \left(I(-t)H(-t)I^{-1}(-t) + i\frac{\partial I(-t)}{\partial t}I^{-1}(-t)\right)I(-t)|\psi(-t)\rangle. \qquad (8)$$

Therefore, for the transformation $I(t)$ to be a symmetry of the system modeled by the Hamiltonian $H(t)$, thus producing an independent solution $I(-t)|\psi(-t)\rangle$ of the Schrödinger equation from a given solution $|\psi(t)\rangle$, we end up with the equation

$$i\frac{\partial I(t)}{\partial t} + H(-t)I(t) - I(t)H(t) = 0, \qquad (9)$$

which defines an anti-linear invariant for a non-Hermitian Hamiltonian. If we had considered a linear instead of antilinear transformation $I(t)$, we would have obtained the equation

$$i\frac{\partial I(t)}{\partial t} + [I(t), H(t)] = 0, \qquad (10)$$

which defines a linear dynamical invariant for a non-Hermitian Hamiltonian $H(t)$. The Eq. (10) is exactly that defining the Lewis & Riesenfeld linear dynamical invariant for a Hermitian Hamiltonian $H(t)$ [43–46].

For a TI symmetry $I$, the Eq. (9) simplifies to the form

$$IH(t)I^{-1} = H(-t), \qquad (11)$$

and for the case where both the symmetry and the Hamiltonian are TI operators, the condition (11) is further simplified to the commutation

$$[I, H] = 0. \qquad (12)$$

Regarding the TI $\mathcal{PT}$ operation, the condition for a TD Hamiltonian to be $\mathcal{PT}$-symmetric is given by

$$\mathcal{PT}H(t)(\mathcal{PT})^{-1} = H(-t), \qquad (13)$$

which reduces, for TI Hamiltonians, to the commutation relation $[\mathcal{PT}, H] = 0$.

We thus verify that the condition for the TD operator $I(t)$ to be the symmetry associated with a TD Hamiltonian $H(t)$, given by the differential equation (9), simplifies to the algebraic equation (11) for a TI symmetry operator. This represents a major reduction in the generality of the symmetry operator, which becomes even greater for a TI Hamiltonian.

Following the reasonings in Ref. [44], where a method for the construction of nonlinear Lewis & Riesenfeld TD invariants is presented, we define the general symmetry operator as the product $I(t) = \Lambda(t)\mathcal{U}(t)$, with $\Lambda(t)$ being either a unitary or nonunitary operator. Regarding $\mathcal{U}(t)$, from now on we assume it to be antilinear in accordance with the condition imposed in

references [4,29], whose results we want to rescue in the scenario of TI Hamiltonian, symmetry and metric operators.

Considering the product $I(t) = \Lambda(t)\mathcal{U}(t)$, Eq. (9) can be rewritten in the form

$$\left(i\frac{\partial \Lambda(t)}{\partial t} + H(-t)\Lambda(t) - \Lambda(t)H(t)\right)\mathcal{U}(t) + \Lambda(t)\left(i\frac{\partial \mathcal{U}(t)}{\partial t} + [H(t),\mathcal{U}(t)]\right) = 0. \qquad (14)$$

By also rewriting the Hamiltonian as $H(t) = H_0(t) + V(t)$, with $H_0(t)$ being either a diagonal or nondiagonal operator with known eigenstates, we propose the ansatz $\mathcal{U}(t) = \mathcal{R}(t)\mathcal{T}$, with $\mathcal{T}$ being the time-reversal operator and $\mathcal{R}(t) = e^{i\phi(t)H_0(t)}$, with a TD complex parameter $\phi(t)$. For a Hermitian $H_0(t)$, $\mathcal{U}(t)$ then becomes an antiunitary operator. We thus define the TD operator

$$\Theta(t) = \left(i\frac{\partial \mathcal{U}(t)}{\partial t} + [H(t),\mathcal{U}(t)]\right)\mathcal{U}^{-1}(t), \qquad (15)$$

such that Eq. (14) becomes

$$i\frac{\partial \Lambda(t)}{\partial t} + H(-t)\Lambda(t) - \Lambda(t)H(t) = -\Lambda(t)\Theta(t). \qquad (16)$$

In summary, to obtain $I(t) = \Lambda(t)\mathcal{U}(t)$, we first compute the TD operator $\Theta(t)$ from Eq. (15), by taking the advantage of the known eigenstate basis of $H_0(t)$ which defines $\mathcal{R}(t)$. Next, starting from an ansatz for $\Lambda(t)$, based on the symmetry group of $V(t)$, we then compute this operator from Eq. (16), what finally gives us the symmetry $I(t)$. It is evidently straightforward to derive the equivalent of Eq. (9) for a linear symmetry operator, with the same ansatz $I(t) = \Lambda(t)\mathcal{U}(t)$ applying for its solution.

## 4 The antilinear symmetry described by a couple of linear and antilinear metric operators

Let us consider a TD non-Hermitian Hamiltonian $H(t)$ which obeys the TD pseudo-Hermiticity relation given by Eq. (2): $H^\dagger(t)\rho(t) - \rho(t)H(t) = i\partial_t\rho(t)$. Starting with the Schrödinger equation for $H(t)$, $i\partial_t|\psi(t)\rangle = H(t)|\psi(t)\rangle$, applying the linear metric operator $\rho(t)$ on its l.h.s., and assuming the relation in Eq. (2), we obtain

$$i\frac{\partial}{\partial t}|\chi(t)\rangle = H^\dagger(t)|\chi(t)\rangle, \qquad (17)$$

where we have defined $|\chi(t)\rangle = \rho(t)|\psi(t)\rangle$. Next, assuming that $H(t)$ also obeys a TD anti-pseudo-Hermiticity relation

$$H^\dagger(t)\Xi(t) - \Xi(t)H(-t) = i\dot{\Xi}(t), \qquad (18)$$

for the TD antilinear metric operator $\Xi(t)$, the application of the operator $\Xi(-t)$ on the l.h.s. of the Schrödinger equation for $H(t)$, leads us again to the Eq. (17) once we define $|\chi(t)\rangle = \Xi(t)|\psi(-t)\rangle$.

Considering both the TD pseudo-Hermiticity relations, in Eqs. (2) and (18), we derive $H^\dagger(t) = \rho(t)H(t)\rho^{-1}(t) + i\dot{\rho}(t)\rho^{-1}(t)$ from the former and then substitute this adjoint Hamiltonian into the latter to obtain

$$i\frac{\partial}{\partial t}\left[\Xi^{-1}(t)\rho(t)\right] = \left[\Xi^{-1}(t)\rho(t)\right]H(t) - H(-t)\left[\Xi^{-1}(t)\rho(t)\right]. \qquad (19)$$

It is straightforward and remarkable to verify that the above expression recovers the Eq. (9) for the TD antilinear symmetry operator defined as

$$I(t) = \Xi^{-1}(t)\rho(t). \tag{20}$$

In fact, for the case where only the Hamiltonian $H(t)$ is a TD operator, we then obtain the simplified linear and antilinear pseudo-Hermiticity relations $H^\dagger(t)\rho = \rho H(t)$ and $H^\dagger(t)\Xi = \Xi H(-t)$, with the TI antilinear symmetry $I = \Xi^{-1}\rho$. When the Hamiltonian is also a TI operator, we then retrieve from our assumption of a TD anti-pseudo-Hermitian relation (18), the results proved by Mostafazadeh in Ref. [4], that *every (non-Hermitian) diagonalizable Hamiltonian is anti-pseudo-Hermitian and that the pseudo-Hermiticity of the Hamiltonian implies the presence of an antilinear symmetry*. In fact, in this case we have $\rho H \rho^{-1} = H^\dagger = \Xi H \Xi^{-1}$, and hence $\left[H, \Xi^{-1}\rho\right] = 0$.

We have thus verified that, for TD Hamiltonian, symmetry and metric operators, we have derived the TD counterpart of the important Mostafazadeh's relation for the symmetry operator, $I = \Xi^{-1}\rho$. We do not, of course, have a counterpart to the theorem proved by Mostafazadeh in the TI scenario, but verifying that the symmetry operator $I(t) = \Lambda(t)\mathcal{U}(t)$ we have derive through Eq. (9) can also be written in the form that generalizes Mostafazadeh's expression to the TD scenario, is significant and will be explored below.

# 5 The TD non-Hermitian Hamiltonian of a cavity field under linear amplification

The TD non-Hermitian Hamiltonian modeling a cavity field under linear amplification is given by

$$H(t) = \omega(t)a^\dagger a + \alpha(t)a + \beta(t)a^\dagger, \tag{21}$$

with the TD parameters $\omega(t)$, $\alpha(t)$, and $\beta(t)$ being complex functions. Here we just demand that $H^\dagger(t) \neq H(t)$, such that $\omega^*(t) \neq \omega(t)$ and/or $\alpha^*(t) \neq \beta(t)$. The usual requirement for the Hamiltonian (21) to be $\mathcal{PT}$-symmetric, given by Eq. (13), imposes the more restrictive conditions $\omega^*(-t) = \omega(t)$, $\alpha^*(-t) = -\alpha(t)$, and $\beta^*(-t) = -\beta(t)$. From Eq. (13) we also verify that the Hamiltonian (21) is $\mathcal{PT}$-symmetric under spatial reflection about both $x_0 = 0$ and

$$x_0 = -\sqrt{\frac{1}{2m\omega(t)}}\frac{\alpha(t) + \beta(t)}{\omega(t)} = \text{real constant}. \tag{22}$$

For the case of a TI Hamiltonian, $x_0 \neq 0$ implies a Hermitian Hamiltonian, whereas for a TD Hamiltonian, $x_0 \neq 0$ imposes constraints on the Hamiltonian parameters which do not occur for $x = 0$.

## 5.1 The TD antilinear symmetry operator

Considering the method proposed for deriving the symmetry operator, we rewrite the Hamiltonian (21) in the form $H(t) = H_0(t) + V(t)$, with $H_0(t) = \omega(t)a^\dagger a$ and $V(t) = \alpha(t)a + \beta(t)a^\dagger$. We then define the operator $\mathcal{R}(t) = e^{-i\phi(t)a^\dagger a}$, such that $\mathcal{U}(t) = e^{-i\phi(t)a^\dagger a}\mathcal{T}$. Consequently, using Eq. (15) we obtain

$$\Theta(t) = \left[\omega(t) - \omega^*(t) + \dot\phi(t)\right]a^\dagger a + \left[\alpha(t) - \alpha^*(t)e^{i\phi(t)}\right]a + \left[\beta(t) - \beta^*(t)e^{-i\phi(t)}\right]a^\dagger, \tag{23}$$

where the dot indicates a time derivative. Next, we consider, as an ansatz, the generalized displacement operator

$$\Lambda(t) = e^{\nu(t)a^\dagger + \lambda(t)a + \mu(t)}, \tag{24}$$

which becomes a unitary operator for $\lambda(t) = -\nu^*(t)$, and a Hermitian operator for $\lambda(t) = \nu^*(t)$. We thus obtain from Eqs. (24) and (16):

$$i\frac{\partial \Lambda(t)}{\partial t} + H(-t)\Lambda(t) - \Lambda(t)H(t) = -\left[A(t)a^\dagger a + B(t)a + C(t)a^\dagger + D(t)\right]\Lambda(t), \qquad (25)$$

where

$$A(t) = \omega(t) - \omega(-t), \qquad (26a)$$

$$B(t) = -\dot{\lambda}(t) + \omega(t)\lambda(t) + \alpha(t) - \alpha(-t), \qquad (26b)$$

$$C(t) = -\dot{\nu}(t) - \omega(t)\nu(t) + \beta(t) - \beta(-t), \qquad (26c)$$

$$D(t) = -\dot{\mu}(t) - \frac{1}{2}\left[\dot{\nu}(t)\lambda(t) - \nu(t)\dot{\lambda}(t)\right]$$
$$- \omega(t)\lambda(t)\nu(t) - \alpha(t)\nu(t) + \beta(t)\lambda(t). \qquad (26d)$$

From the r.h.s. of Eqs. (16) and (25), it follows that

$$\Lambda(t)\Theta(t)\Lambda^{-1}(t) = A(t)a^\dagger a + B(t)a + C(t)a^\dagger + D(t), \qquad (27)$$

and by substituting Eqs. (23) and (24) in Eq. (27), we obtain

$$\phi(t) = \phi_0 + \int_0^t \left[\omega^*(\tau) - \omega(-\tau)\right]d\tau, \qquad (28a)$$

$$\dot{\lambda}(t) = \omega(-t)\lambda(t) + \alpha^*(t)e^{i\phi(t)} - \alpha(-t), \qquad (28b)$$

$$\dot{\nu}(t) = -\omega(-t)\nu(t) + \beta^*(t)e^{-i\phi} - \beta(-t), \qquad (28c)$$

$$\mu(t) = \mu_0 - \frac{1}{2}\int_0^t \left\{\left[\alpha^*(\tau)e^{i\phi(\tau)} + \alpha(-\tau)\right]\nu(\tau)\right.$$
$$\left. - \left[\beta^*(\tau)e^{-i\phi(\tau)} + \beta(-\tau)\right]\lambda(\tau)\right\}d\tau. \qquad (28d)$$

Note from Eq. (28d) that the parameter $\mu(t)$ is added to the generalized displacement operator to avoid undesirable constraints in the Hamiltonian's parameters. For the particular case of a unitary operator $\Lambda(t)$, where $\lambda(t) = -\nu^*(t)$, we use Eqs. (28b) and (28c) to obtain

$$\nu(t) = \frac{\alpha(t) + \beta^*(t)}{\omega(-t) + \omega^*(-t)}e^{-i\phi} - \frac{\alpha^*(-t) + \beta(-t)}{\omega(-t) + \omega^*(-t)}. \qquad (29)$$

Therefore, from Eqs. (28) we obtain the parameters defining the TD antilinear symmetry operator

$$I(t) = \mathcal{D}(t)\mathcal{R}(t)\mathcal{T}, \qquad (30)$$

where we have replaced $\Lambda$ for $\mathcal{D}$, which, for a unitary $\Lambda$ becomes the displacement operator. This symmetry operator describes the successive actions of a time-reversal operator $\mathcal{T}$, a TD global rotation in phase space $\mathcal{R}(t) = e^{-i\phi(t)a^\dagger a}$ and, finally, let us say, a TD generalized displacement in phase space $\mathcal{D}(t) = e^{\nu(t)a^\dagger + \lambda(t)a + \mu(t)}$. For a unitary $\Lambda$, this TD symmetry $I(t) = \mathcal{D}(t)\mathcal{R}(t)\mathcal{T}$ resembles the evolution operator for the Hermitized counterpart of the TD Hamiltonian in Eq. (21), except, of course, for the time-reversal operation. Such evolution operator can be derived following the reasonings in Refs. [12, 25, 26, 44–46]. Therefore, if applied to a given state of the Hermitized counterpart of our Hamiltonian, this peculiar TD symmetry operator $I(t) = \mathcal{D}(t)\mathcal{R}(t)\mathcal{T}$ causes the probability distribution to trace an upward spiral in phase space.

## 5.2 The Dyson map and pseudo-Hermiticity relation

For treating a TD non-Hermitian Hamiltonian we consider a TD Dyson map $\eta$ which results, in general [24], in a TD metric operator $\rho = \eta^\dagger \eta$. Otherwise, the TD pseudo-Hermiticity relation (2) imposes undesirable constraints on the TD parameters of the Hamiltonian. For the TD Dyson map we consider the ansatz

$$\eta = e^{\epsilon a^\dagger a + \gamma a + \gamma^* a^\dagger}, \tag{31}$$

with $\epsilon(t)$ being a real function. To determine its time derivative we use the method of parameter differentiation [47], by which

$$\frac{\partial}{\partial t} e^Z = \int_0^1 e^{xZ} \frac{\partial Z}{\partial t} e^{-xZ} dx \, e^Z, \tag{32}$$

where $Z = \epsilon a^\dagger a + \gamma a + \gamma^* a^\dagger$. We thus obtain the Hamiltonian

$$h = W a^\dagger a + U a + V a^\dagger + F, \tag{33}$$

where

$$W = i\dot{\epsilon} + \omega, \tag{34a}$$

$$U = i\dot{\gamma} + i\left(1 - \frac{1 - e^{-\epsilon}}{\epsilon}\right)\left(\frac{\gamma}{\epsilon}\dot{\epsilon} - \dot{\gamma}\right) + \omega\gamma\frac{1 - e^{-\epsilon}}{\epsilon} + \alpha e^{-\epsilon}, \tag{34b}$$

$$V = i\dot{\gamma}^* + i\left(1 + \frac{1 - e^{\epsilon}}{\epsilon}\right)\left(\frac{\gamma^*}{\epsilon}\dot{\epsilon} - \dot{\gamma}^*\right) + \omega\gamma^*\frac{1 - e^{\epsilon}}{\epsilon} + \beta e^{\epsilon}, \tag{34c}$$

$$F = 2|\gamma|^2\left(i\frac{\dot{\epsilon}}{\epsilon} + \omega\right)\frac{1 - \cosh\epsilon}{\epsilon^2} - \frac{i}{\epsilon}\left(1 - \frac{1 - e^{-\epsilon}}{\epsilon}\right)\gamma^*\dot{\gamma}$$
$$- \frac{i}{\epsilon}\left(1 + \frac{1 - e^{\epsilon}}{\epsilon}\right)\gamma\dot{\gamma}^* - \frac{\alpha\gamma^*}{\epsilon}\left(1 - e^{-\epsilon}\right) - \frac{\beta\gamma}{\epsilon}\left(1 - e^{\epsilon}\right). \tag{34d}$$

To ensure the Hermiticity of $h(t)$ we impose a complex TD frequency $\omega(t) = \omega_R(t) - i\dot{\epsilon}(t)$, with $\omega_R(t)$ being a real function, in addition to $U = V^*$ and $F \in \mathbb{R}$, what demands that

$$\dot{\gamma} + \left(\frac{\epsilon\coth\epsilon - 1}{\epsilon}\dot{\epsilon} - i\omega_R\right)\gamma - i\frac{\epsilon}{2\sinh\epsilon}\left(\alpha e^{-\epsilon} - \beta^* e^{\epsilon}\right) = 0, \tag{35a}$$

$$2i\left(\frac{\sinh\epsilon}{\epsilon} - 1\right)(\gamma\dot{\gamma}^* + \gamma^*\dot{\gamma}) + 2\left(\omega - \omega^* + 2i\frac{\dot{\epsilon}}{\epsilon}\right)|\gamma|^2\frac{1 - \cosh\epsilon}{\epsilon}$$
$$- (\alpha\gamma^* - \alpha^*\gamma)\left(1 - e^{-\epsilon}\right) - (\beta\gamma - \beta^*\gamma^*)(1 - e^{\epsilon}) = 0. \tag{35b}$$

From (35a) we obtain

$$\gamma = e^{-\chi}\left(\gamma_0 + i\int_0^t \frac{\epsilon e^{\chi}}{2\sinh\epsilon}\left(\alpha e^{-\epsilon} - \beta^* e^{\epsilon}\right) d\tau\right), \tag{36}$$

with

$$\chi = \int_0^t \left(\frac{(\epsilon\coth\epsilon - 1)}{\epsilon}\dot{\epsilon} - i\omega_R\right) d\tau. \tag{37}$$

Now, by substituting Eq. (35a) into Eq. (35b), and admitting momentarily the approximation $\epsilon \ll 1$, we obtain

$$\epsilon \simeq \exp\left(-i\int_0^t \frac{(\alpha^* + \beta)\gamma - (\alpha + \beta^*)\gamma^*}{2|\gamma|^2} d\tau\right), \tag{38}$$

showing that the Hermiticity requirements in Eqs. (35) imposes no additional constraints on the Hamiltonian parameters, apart from the complex TD frequency $\omega(t) = \omega_R(t) - i\dot{\epsilon}(t)$ coming from Eq. (34a). Otherwise, when we assume that $\omega(t)$ is real from Eq. (21), it follows that $\epsilon$ must be constant, Which leads to a new Hamiltonian $h$ in Eq. (33), and consequently to a new system in Eq. (34) and a new hermitization condition in Eq. (35). When the simplified Dyson map $\eta = e^{\gamma a + \gamma^* a^\dagger}$ is considered, as in Ref. [11], with $\epsilon = 0$, the pseudo-Hermiticity requirement of a complex frequency simplifies to that of a real one, $\omega(t) = \omega_R(t)$, still with no constraints on $\alpha(t)$ and $\beta(t)$.

Therefore, when considering the symmetry operator for the TD pseudo-Hermitian Hamiltonian (33), with $\gamma$ and $\epsilon$ following from Eqs. (36) and (38), we must necessarily assume the function $\omega(t)$ appearing in Eqs. (28) to be of the form $\omega(t) = \omega_R(t) - i\dot{\epsilon}(t)$ (or $\omega = \omega_R$ for the particular Dyson map $\eta = e^{\gamma a + \gamma^* a^\dagger}$). Despite of the frequency constraint, the symmetry operator in Eq. (30), have no restrictions for the amplification parameters $\alpha(t)$ and $\beta(t)$.

### 5.3 From $I(t)$ in Eq. (30) to $\mathcal{PT}$

The TI $\mathcal{PT}$ operator can be directly recovered from Eq. (30), starting with the constraints under which the Hamiltonian (21) is $\mathcal{PT}$-symmetric: $\omega^*(-t) = \omega(t)$, $\alpha^*(-t) = -\alpha(t)$, and $\beta^*(-t) = -\beta(t)$. Assuming also a unitary $\Lambda(t)$, we verify from Eqs. (28a) and (28b) that the rotation is reduced to a TI operator with $\phi(t) = \phi_0$, while the parameter $\nu(t)$ of the displacement operator is simplified to

$$\nu(t) = \frac{\alpha(t) + \beta^*(t)}{\omega^*(t) + \omega(t)} \left(1 + e^{i\phi_0}\right). \tag{39}$$

For a TI symmetry operator, where $I = \mathcal{DU}$ must be a TI parameter, we must then assume, to avoid undesirable constraints on the Hamiltonian parameters, that $\phi_0 = (2n+1)\pi$, with $n \in \mathbb{Z}$. From this assumption we obtain $\nu(t) = \mu(t) = 0$, and noting that the parity operator can be written in the form $e^{-i(2n+1)\pi a^\dagger a}$, we finally recover the TI operator $\mathcal{PT}$ from Eq. (30), i.e.,:

$$I(t) \rightarrow I = e^{-i(2n+1)\pi a^\dagger a}\mathcal{T} = \mathcal{PT}. \tag{40}$$

## 6 The TI non-Hermitian Hamiltonian of a cavity field under linear amplification

Now we consider the particular case of a TI non-Hermitian Hamiltonian

$$H = \omega a^\dagger a + \alpha a + \beta a^\dagger, \tag{41}$$

with $\omega^* \neq \omega$ and/or $\alpha^* \neq \beta$. The $\mathcal{PT}$-symmetry of $H$, now following from the commutation $[\mathcal{PT}, H] = 0$, imposes here the more restrictive conditions $\omega^* = \omega$, $\alpha^* = -\alpha$, and $\beta^* = -\beta$, and enables the spatial reflection only about $x_0 = 0$, for

$$\omega \in \mathbb{R}, \quad \alpha = |\alpha| e^{i(n+1/2)\pi}, \quad \beta = |\beta| e^{i(m+1/2)\pi}, \quad \text{with } n, m \in \mathbb{Z}. \tag{42}$$

The case $x_0 \neq 0$ implies a Hermitian Hamiltonian as anticipated above.

### 6.1 The TI antilinear symmetry operator

The condition for the TI Hamiltonian (41) to be invariant under a TI antilinear operator $I$ is given by the commutation relation $[I, H] = 0$. From the knowledge of the TD symmetry

operator in Eq. (30), it is natural to assume for its TI equivalent the form $I = \mathcal{D}\mathcal{R}\mathcal{T}$, with a TI global rotation $\mathcal{R} = e^{-i\phi a^\dagger a}$ and a TI $\mathcal{D} = e^{\nu a^\dagger + \lambda a}$. We have neglected the parameter $\mu$ added to the Eq. (24) since it is insensitive to the commutation relation $[I, H] = 0$, which imposes the equations

$$\omega^* = \omega, \tag{43a}$$

$$\omega\lambda - \alpha + \alpha^* e^{i\phi} = 0, \tag{43b}$$

$$\omega\nu + \beta - \beta^* e^{-i\phi} = 0, \tag{43c}$$

$$\omega\lambda\nu + \alpha^* \nu e^{i\phi} - \beta^* \lambda e^{-i\phi} = 0. \tag{43d}$$

For a unitary $\Lambda$ ($\lambda = -\nu^*$), and using the polar forms $\alpha = |\alpha| e^{i\varphi_\alpha}$ and $\beta = |\beta| e^{i\varphi_\beta}$, it follows from Eqs. (43) that

$$\phi = i \ln \frac{\alpha^* - \beta}{\alpha - \beta^*}, \tag{44a}$$

$$|\nu| = \frac{1}{\omega}\left[|\alpha|\sin\left(\varphi_\alpha - \frac{\phi}{2}\right) - |\beta|\sin\left(\varphi_\beta + \frac{\phi}{2}\right)\right], \tag{44b}$$

$$\varphi_\nu = \left(n + \frac{1}{2}\right)\pi - \frac{\phi}{2}. \tag{44c}$$

Substituting Eqs. (44b) and (44c) into Eq. (43d) (with $\lambda = -\nu^*$), we obtain the expression

$$|\nu|\left[|\alpha|\cos\left(\varphi_\alpha - \frac{\phi}{2}\right) - |\beta|\cos\left(\varphi_\beta + \frac{\phi}{2}\right)\right] = 0, \tag{45}$$

which result in two different solutions, one for $|\nu| = 0$ and the other for $|\nu| \neq 0$. For $|\nu| = 0$, the Eq. (45) is automatically satisfied and using Eq. (44b) we obtain the constraints $\varphi_\alpha = \phi/2 + n\pi$ and $\varphi_\beta = -\phi/2 + m\pi$, with $n, m \in \mathbb{Z}$, which satisfy Eq. (44a) for $n = 0$, such that $\phi = 2\varphi_\alpha$ and the Hamiltonian's parameters become

$$\omega \in \mathbb{R}, \quad \alpha = |\alpha| e^{i\varphi_\alpha}, \quad \beta = |\beta| e^{i[m\pi - \varphi_\alpha]}, \tag{46}$$

and the TI antiunitary symmetry operator

$$I = e^{-2i\varphi_\alpha a^\dagger a}\mathcal{T}. \tag{47}$$

For $|\nu| \neq 0$, we get the constraint

$$\frac{\cos(\varphi_\alpha - \phi/2)}{\cos(\varphi_\beta + \phi/2)} = \frac{|\beta|}{|\alpha|} = \mathfrak{p}, \tag{48}$$

which, considering $\varphi_\alpha + \varphi_\beta = \varphi$, leads to the relation

$$\phi = 2\varphi_\alpha - 2\tan^{-1}\frac{1 - \mathfrak{p}\cos\varphi}{\mathfrak{p}\sin\varphi}, \tag{49}$$

in agreement with Eq. (44a). From the above results, we obtain from Eq. (44b) the expression

$$|\nu| = \frac{|\alpha|}{\omega}\frac{1 - \mathfrak{p}^2}{1 + \mathfrak{p}^2 - 2\mathfrak{p}\cos\varphi}, \tag{50}$$

which imposes

$$\mathfrak{f} = \mathfrak{p}^2 - \frac{2\omega\cos\varphi}{\omega + |\alpha|}\mathfrak{p} + \frac{\omega - |\alpha|}{\omega + |\alpha|} < 0, \tag{51}$$

together with $|\cos\varphi| > \sqrt{\omega^2 - |\alpha|^2}/2\omega$, and then $\mathfrak{p}_- < \mathfrak{f} < \mathfrak{p}_+$, with

$$\mathfrak{p}_\pm = \frac{\omega}{\omega + |\alpha|}\left(1 \pm \sqrt{1 - \frac{\omega^2 - |\alpha|^2}{4\omega^2\cos^2\varphi}}\right). \tag{52}$$

The parameters in Eqs. (49) and (50), under the above constraints for $\varphi$ and $\mathfrak{f}$, define the TI antiunitary symmetry operator

$$I = e^{\nu a^\dagger - \nu^* a}e^{-i\phi a^\dagger a}\mathcal{T}. \tag{53}$$

Although the TI continuous symmetry operators in Eqs. (47) and (53) are particular cases of the TD symmetry operator in Eq. (30), they are generalizations of the discrete parity and time-reversal transformation. Differently from the operator $I(t)$ in Eq. (30), whose TD parameters depend on the Hermiticity conditions only through the frequency requirement $\omega = \omega_R - i\dot\epsilon$ (or $\omega = \omega_R$ for the particular Dyson map $\eta = e^{\gamma a + \gamma^* a^\dagger}$), the operators in Eqs. (47) and (53) takes into account the constraints imposed on $\omega$, $\alpha$ and $\beta$. The TI non-Hermitian Hamiltonians and consequently the associated symmetry operators are more vulnerable than their general TD equivalents to the constraints imposed by the pseudo-Hermiticity relation. This vulnerability to the constraints follows from the more stringent condition for the invariance of a TI Hamiltonian: $[I, H] = 0$.

### 6.1.1 Bender-Berry-Mandilara

Although it is straightforward to verify the validity of the relation $I^{2k} = 1$, with $k$ odd [29], for the symmetry operator in Eq. (47), its validity for the operator in Eq. (53) demands a little algebra. In fact, for the operator in Eq. (53) we obtain

$$\begin{aligned}I^2 &= e^{\nu a^\dagger - \nu^* a}e^{-i\phi a^\dagger a}e^{\nu^* a^\dagger - \nu a}e^{i\phi a^\dagger a}\mathcal{T}^2 \\ &= \exp\left(\nu a^\dagger - \nu^* a\right)\exp\left(\nu^* e^{-i\phi}a^\dagger - \nu e^{i\phi}a\right) = 1,\end{aligned} \tag{54}$$

since it follows from Eqs. (44c) and (50) that $\nu e^{i\phi} = -\nu^*$.

It is important to note that the TD symmetry operator $I(t)$ does not obey the Bender-Berry-Mandilara relation. Although we still have the relation

$$I^2(t) = \exp\left[\nu(t)a^\dagger - \nu^*(t)a\right]\exp\left[\nu^*(t)e^{-i\phi(t)}a^\dagger - \nu(t)e^{i\phi(t)}a\right], \tag{55}$$

the equality $\nu(t)e^{i\phi(t)} = -\nu^*(t)$ is no longer satisfied in the TD scenario.

## 6.2 Dyson map and pseudo-Hermiticity

For the TI Hamiltonian (41), we consider the TI Dyson map $\eta = e^{\epsilon a^\dagger a + \gamma a + \gamma^* a^\dagger}$, with $\epsilon \in \mathbb{R}$, leading to the Dyson relation (6)

$$h = \eta H \eta^{-1} = \omega a^\dagger a + u a + v a^\dagger + f, \tag{56}$$

where $u$, $v$ and $f$ follow directly from $U$, $V$ and $F$ in Eqs. (34). The Hermiticity condition $h = h^\dagger$ imposes $\omega, f \in \mathbb{R}$ and $u = v^*$, such that

$$\gamma = \frac{\epsilon}{\omega}\frac{\alpha e^{-\epsilon} - \beta^* e^\epsilon}{e^{-\epsilon} - e^\epsilon}, \tag{57a}$$

$$\frac{\gamma^*}{\gamma} = \frac{\alpha^*\left(1 - e^{-\epsilon}\right) + \beta\left(e^\epsilon - 1\right)}{\alpha\left(1 - e^{-\epsilon}\right) + \beta^*\left(e^\epsilon - 1\right)}. \tag{57b}$$

By substituting Eq. (57a) into Eq. (57b) we obtain $\beta\alpha \in \mathbb{R}$. With the polar forms $\alpha = |\alpha|e^{i\varphi_\alpha}$ and $\beta = |\beta|e^{i\varphi_\beta}$, it follows that $\varphi_\beta = n\pi - \varphi_\alpha$, with $n \in \mathbb{Z}$. Therefore, for the chosen TI Dyson map $\eta$, the Hamiltonian (41) becomes pseudo-Hermitian, with $\gamma$ given by Eq. (57a) and $\epsilon$ being a free real parameter, under the constraints

$$\omega \in \mathbb{R}, \quad \alpha = |\alpha|e^{i\varphi_\alpha}, \quad \beta = |\beta|e^{i(n\pi-\varphi_\alpha)}, \tag{58}$$

exactly those in Eq. (46). Therefore, for the case $|\nu| = 0$, the pseudo-Hermiticity does not impose additional constraints on the symmetry operator (47) beyond those already following from the commutation relation $[I, H] = 0$. The same does not apply to the case $|\nu| \neq 0$ which leads to the much more complex symmetry operator (53).

## 6.3 From $I$ in Eq. (47) to $\mathcal{PT}$

Under the requirement for the $\mathcal{PT}$-symmetry invariance of the TI Hamiltonian (41), which imposes $\varphi_\alpha = \varphi_\beta = (n + 1/2)\pi$, the TI symmetry in Eq. (47) automatically reduces to

$$I = e^{-i(2n+1)\pi a^\dagger a}\mathcal{T} = \mathcal{PT}. \tag{59}$$

# 7 The TD non-Hermitian Hamiltonian of a cavity field under parametric amplification

The TD non-Hermitian Hamiltonian for a cavity field under parametric amplification is given by

$$H = \omega(t)\left(a^\dagger a + 1/2\right) + \alpha(t)a^2 + \beta(t)a^{\dagger 2}, \tag{60}$$

with complex TD parameters $\omega(t)$, $\alpha(t)$, and $\beta(t)$. We only require this Hamiltonian to be non-Hermitian, $H^\dagger(t) \neq H(t)$, such that $\omega^*(t) \neq \omega(t)$ and/or $\alpha^*(t) \neq \beta(t)$. The usual $\mathcal{PT}$-symmetry requirement for Hamiltonian (60) imposes the more restrictive conditions $\omega^*(-t) = \omega(t)$, $\alpha^*(-t) = \alpha(t)$, and $\beta^*(-t) = \beta(t)$.

## 7.1 The TD antilinear symmetry operator

Rewriting the Hamiltonian (60) in the form $H(t) = H_0(t) + V(t)$, with $H_0(t) = \omega(t)a^\dagger a$ and $V(t) = \alpha(t)a^2 + \beta(t)a^{\dagger 2}$, we then define $\mathcal{R}(t) = e^{-i\phi(t)a^\dagger a}$ and $\mathcal{U}(t) = e^{-i\phi(t)a^\dagger a}\mathcal{T}$, enabling us to derive from Eq. (15) the operator

$$\Theta = \left(\dot{\phi} + \omega - \omega^*\right)\left(a^\dagger a + 1/2\right) + \left(\alpha - \alpha^* e^{2i\phi}\right)a^2 + \left(\beta - \beta^* e^{-2i\phi}\right)a^{\dagger 2}. \tag{61}$$

We now consider the ansatz

$$\Lambda(t) = e^{\xi(t)a^2 + \zeta(t)a^{\dagger 2}}, \tag{62}$$

which becomes a unitary operator for $\zeta(t) = -\xi^*(t)$, and an Hermitian operator for $\zeta(t) = \xi^*(t)$. From the ansatz in Eq. (62) we compute the l.h.s. of Eq. (16), giving us

$$i\hbar\frac{\partial\Lambda(t)}{\partial t} + H(-t)\Lambda(t) - \Lambda(t)H(t) = \left[A(t)\left(a^\dagger a + 1/2\right) + B(t)a^2 + C(t)a^{\dagger 2}\right]\Lambda(t), \tag{63}$$

where we have defined

$$
\begin{aligned}
A(t) = {}&\omega(-t) - \omega(t)\cos\left(4\sqrt{\xi\zeta}\right) + \left(\alpha(t)\sqrt{\xi^{-1}\zeta} - \beta(t)\sqrt{\xi\zeta^{-1}}\right)\sin\left(4\sqrt{\xi\zeta}\right) \\
&+ i\frac{\left(\dot{\xi}\zeta - \xi\dot{\zeta}\right)}{4\xi\zeta}\left[1 - 8\xi\zeta - \cos\left(4\sqrt{\xi\zeta}\right)\right],
\end{aligned}
\tag{64a}
$$

$$
\begin{aligned}
B(t) = {}&-\frac{1}{2}\omega(t)\sqrt{\xi\zeta^{-1}}\sin\left(4\sqrt{\xi\zeta}\right) + \alpha(-t) - \alpha(t)\cos^2\left(2\sqrt{\xi\zeta}\right) \\
&- \beta(t)\xi\zeta^{-1}\sin^2\left(2\sqrt{\xi\zeta}\right) + i\dot{\xi} + 2i\frac{\xi\left(\dot{\xi}\zeta - \xi\dot{\zeta}\right)}{\sqrt{\xi\zeta}}\left[4\sqrt{\xi\zeta} - \sin\left(4\sqrt{\xi\zeta}\right)\right],
\end{aligned}
\tag{64b}
$$

$$
\begin{aligned}
C(t) = {}&\frac{1}{2}\omega(t)\sqrt{\xi^{-1}\zeta}\sin\left(4\sqrt{\xi\zeta}\right) + \beta(-t) - \beta(t)\cos^2\left(2\sqrt{\xi\zeta}\right) \\
&- \alpha(t)\xi^{-1}\zeta\sin^2\left(2\sqrt{\xi\zeta}\right) + i\dot{\zeta} - 2i\frac{\left(\dot{\xi}\zeta - \xi\dot{\zeta}\right)\zeta}{\sqrt{\xi\zeta}}\left[4\sqrt{\xi\zeta} - \sin\left(4\sqrt{\xi\zeta}\right)\right].
\end{aligned}
\tag{64c}
$$

Next, we substitute Eq. (63) into Eq. (16) to obtain

$$
\Lambda(t)\Theta(t)\Lambda^{-1}(t) = -\left[A(t)\left(a^\dagger a + \frac{1}{2}\right) + B(t)a^2 + C(t)a^{\dagger 2}\right].
\tag{65}
$$

Finally, by substituting Eqs. (61) and (62) into Eq. (65), we derive the system

$$
\begin{aligned}
&\dot{\phi} + i\frac{\left(\dot{\xi}\zeta - \xi\dot{\zeta}\right)}{4\xi\zeta}\left[1 - 8\xi\zeta - \cos\left(4\sqrt{\xi\zeta}\right)\right]\sec\left(4\sqrt{\xi\zeta}\right) \\
&= \omega^*(t) - \omega(-t)\sec\left(4\sqrt{\xi\zeta}\right) - \left[\alpha^*(t)e^{2i\phi(t)}\sqrt{\xi^{-1}\zeta} - \beta^*(t)e^{-2i\phi(t)}\sqrt{\xi\zeta^{-1}}\right]\tan\left(4\sqrt{\xi\zeta}\right),
\end{aligned}
\tag{66a}
$$

$$
\begin{aligned}
&\dot{\phi} + 2i\dot{\xi}\sqrt{\xi^{-1}\zeta}\csc\left(4\sqrt{\xi\zeta}\right) + 4i\left(\dot{\xi}\zeta - \xi\dot{\zeta}\right)\left[4\sqrt{\xi\zeta}\csc\left(4\sqrt{\xi\zeta}\right) - 1\right] \\
&= \omega^*(t) - 2\alpha(-t)\sqrt{\xi^{-1}\zeta}\csc\left(4\sqrt{\xi\zeta}\right) + \alpha^*(t)e^{2i\phi(t)}\sqrt{\xi^{-1}\zeta}\cot\left(2\sqrt{\xi\zeta}\right) \\
&\quad + \beta^*(t)e^{-2i\phi(t)}\sqrt{\xi\zeta^{-1}}\tan\left(2\sqrt{\xi\zeta}\right),
\end{aligned}
\tag{66b}
$$

$$
\begin{aligned}
&\dot{\phi} - 2i\dot{\zeta}\sqrt{\xi\zeta^{-1}}\csc(4\sqrt{\xi\zeta}) + 4i\left(\dot{\xi}\zeta - \xi\dot{\zeta}\right)\left[4\sqrt{\xi\zeta}\csc(4\sqrt{\xi\zeta}) - 1\right] \\
&= \omega^*(t) + 2\beta(-t)\sqrt{\xi\zeta^{-1}}\csc\left(4\sqrt{\xi\zeta}\right) - \alpha^*(t)e^{2i\phi(t)}\sqrt{\xi^{-1}\zeta}\tan\left(2\sqrt{\xi\zeta}\right) \\
&\quad - \beta^*(t)e^{-2i\phi(t)}\sqrt{\xi\zeta^{-1}}\cot\left(2\sqrt{\xi\zeta}\right),
\end{aligned}
\tag{66c}
$$

from which we compute the variables $\phi(t)$, $\xi(t) = |\xi(t)|\,e^{i\varphi_\xi(t)}$ and $\zeta(t) = |\zeta(t)|\,e^{i\varphi_\zeta(t)}$ which define the operators $\mathcal{U}(t)$ and $\Lambda(t)$. In order to find a possible solution for the system (66), we assume small amplification parameters, $|\xi| \ll 1$ and $|\zeta| \ll 1$, in order to disregard their second order corrections, leaving us with the equations

$$
\dot{\phi}(t) - \omega^*(t) + \omega(-t) + 4\left[\alpha^*(t)\zeta(t)e^{2i\phi(t)} - \beta^*(t)\xi(t)e^{-2i\phi(t)}\right] = 0,
\tag{67a}
$$

$$
2\xi(t)\dot{\phi}(t) + i\dot{\xi}(t) - 2\omega^*(t)\xi(t) - \alpha^*(t)e^{2i\phi(t)} + \alpha(-t) = 0,
\tag{67b}
$$

$$
-2\zeta(t)\dot{\phi}(t) + i\dot{\zeta}(t) + 2\omega^*(t)\zeta(t) - \beta^*(t)e^{-2i\phi(t)} + \beta(-t) = 0.
\tag{67c}
$$

For the particular case where $\Lambda(t)$ is a unitary operator, i.e., $\zeta(t) = -\xi^*(t)$, and $\operatorname{Im}\omega(-t) \neq 0$, the above system simplifies to the decoupled equations

$$
\dot{\phi}(t) = \omega^*(t) - \omega(-t) + 4\left[\alpha^*(t)\xi^*(t)e^{2i\phi(t)} + \beta^*(t)\xi(t)e^{-2i\phi(t)}\right],
\tag{68a}
$$

$$
\xi(t) = \frac{\alpha(-t) - \beta^*(-t) - \left[\alpha^*(t) - \beta(t)\right]e^{2i\phi(t)}}{4i\operatorname{Im}\omega(-t)}.
\tag{68b}
$$

For the case where $\operatorname{Im}\omega(-t) = 0$, the parameter $\xi(t)$ follows from

$$2i\dot{\xi}(t) = 4[\operatorname{Re}\omega(-t)]\xi(t) + [\alpha^*(t) + \beta(t)]e^{2i\phi(t)} - \alpha(-t) - \beta^*(-t). \tag{69}$$

Therefore, from Eqs. (67) we obtain the parameters defining the TD antilinear symmetry operator

$$I(t) = \mathcal{S}(t)\mathcal{R}(t)\mathcal{T}, \tag{70}$$

which becomes an antiunitary operator, with $\Lambda(t)$ being the squeezing operator $\mathcal{S}(t) = e^{\xi(t)a^{\dagger 2} - \xi^*(t)a^2}$ when considering the parameters following from Eqs. (68). This symmetry operator describes the successive action of a time-reversal operator $\mathcal{T}$, a TD global rotation in phase space $\mathcal{R}(t) = e^{-i\phi(t)a^{\dagger}a}$ and a TD operation performed by the generalized squeezing $\mathcal{S}(t) = e^{\xi(t)a^{\dagger 2} + \zeta(t)a^2}$. From Eq. (68b) we verify that the approximation $|\xi| \ll 1$ corresponds to small amplification parameters $|\alpha|$ and $|\beta|$ compared to $|\omega|$.

Similarly to the symmetry operator derived above for the TD non-Hermitian linear Hamiltonian, for the case unitary $\Lambda$ the operator in Eq. (47) resembles the evolution operator for the Hermitized counterpart of the TD Hamiltonian in Eq. (60) [12,25,26,44–46], apart from the time-reversal operation. If applied to a given state of the Hermitized counterpart of our Hamiltonian, this symmetry operator moves the probability distribution across an upward spiral in phase space, continuously squeezing it.

## 7.2 Dyson map and pseudo-Hermiticity

For the TD quadratic Hamiltonian (60) we consider the TD quadratic Dyson map

$$\eta(t) = \exp\left[\epsilon(t)a^{\dagger}a + \gamma(t)a^2 + \gamma^*(t)\left(a^{\dagger}\right)^2\right], \tag{71}$$

which is a positive Hermitian operator under the constraint $\theta^2 = \epsilon^2 - 4|\gamma|^2 > 0$. This map provides us with the relations

$$\eta a\eta^{-1} = \left(\cosh\theta - \frac{\epsilon}{\theta}\sinh\theta\right)a - \frac{2\gamma^*}{\theta}\sinh\theta a^{\dagger}, \tag{72a}$$

$$\eta a^{\dagger}\eta^{-1} = \left(\cosh\theta + \frac{\epsilon}{\theta}\sinh\theta\right)a^{\dagger} + \frac{2\gamma}{\theta}\sinh\theta a, \tag{72b}$$

and consequently with the transformed Hamiltonian

$$h = \eta H\eta^{-1} + i\dot{\eta}\eta^{-1}$$
$$= W\left(a^{\dagger}a + \frac{1}{2}\right) + Ua^2 + Va^{\dagger 2}, \tag{73}$$

where, after defining the parameters

$$\lambda = \frac{2\gamma^*\sinh\theta}{\theta\cosh\theta - \epsilon\sinh\theta}, \tag{74a}$$

$$\lambda_0 = \frac{\theta^2}{[\theta\cosh\theta - \epsilon\sinh\theta]^2}, \tag{74b}$$

$$\chi = |\lambda|^2 - \lambda_0, \tag{74c}$$

we obtain

$$W = -\frac{1}{\lambda_0}\left[\omega\left(\chi + |\lambda|^2\right) + 2(\alpha\lambda + \beta\chi\lambda^*) - \frac{i}{2}\left(\dot{\lambda}_0 - 2\Lambda\dot{\lambda}^*\right)\right], \tag{75a}$$

$$U = \frac{1}{\lambda_0}\left(\omega\lambda^* + \alpha + \beta\lambda^{*2} + \frac{i}{2}\dot{\lambda}^*\right), \tag{75b}$$

$$V = \frac{1}{\lambda_0}\left[\omega\chi\lambda + \alpha\lambda^2 + \beta\chi^2 + \frac{i}{2}\left(\lambda_0\dot{\lambda} + \lambda^2\dot{\lambda}^* - \Lambda\dot{\lambda}_0\right)\right]. \tag{75c}$$

For $h$ in Eq. (73) to be Hermitian we require the constraints $W = W^*$ and $V = U^*$, which results in the system

$$|\dot{z}| = -|z|^2 \left\{ |\omega| \frac{\chi + \Phi^2}{\Phi} \sin\varphi_\omega - 2\left[|\alpha| \sin(\varphi - \varphi_\alpha) - |\beta| \chi \sin(\varphi + \varphi_\beta)\right] \right\} + \frac{|z|}{\Phi}\dot{\Phi}, \quad (76a)$$

$$\dot{\Phi} = \frac{2}{\chi - 1} \left\{ \left[|\omega| \Phi \sin\varphi_\omega - |\alpha| \sin(\varphi - \varphi_\alpha)\right]\left(1 - \Phi^2\right) \right.$$
$$\left. - |\beta| \left[(2\chi - 1)\Phi^2 - \chi^2\right] \sin(\varphi + \varphi_\beta) \right\}, \quad (76b)$$

$$\dot{\varphi} = \frac{2}{(1-\chi)\Phi} \left[|\alpha|\left(1 - \Phi^2\right)\cos(\varphi - \varphi_\alpha) + |\beta|\left(\Phi^2 - \chi^2\right)\cos(\varphi + \varphi_\beta)\right] + 2|\omega|\cos\varphi_\omega, \quad (76c)$$

where we have defined $\gamma = |\gamma| e^{i\varphi}$, $\Phi = \lambda e^{i\varphi}$, and

$$z = \frac{2\gamma}{\epsilon} = |z| e^{i\varphi}, \quad (77)$$

with the initial values $|z(0)|$ and $\varphi(0)$ being two free parameters defining the Dyson Map. After deriving $z$, $\lambda$ and $\varphi$ from the system (76), we the compute $\epsilon$ from the relation

$$\epsilon = \frac{1}{2\sqrt{1-|z|^2}} \ln\left[ \frac{\left(1 + \sqrt{1-|z|^2}\right)\Phi + |z|}{\left(1 - \sqrt{1-|z|^2}\right)\Phi + |z|} \right]. \quad (78)$$

As follows from the system (76), the requirement for the Hamiltonian $h$ to be Hermitian does not imply any restrictions on the Hamiltonian parameters $\omega(t)$, $\alpha(t)$ and $\beta(t)$, differently from what happens with the TD pseudo-Hermitian linear Hamiltonian, where a constraint is required on the frequency of the oscillator. This desired absence of constraints on the parameters defining the TD non-Hermitian quadratic Hamiltonian comes certainly from its SU(1,1) dynamical symmetry.

### 7.3 All-creation and all-annihilation Hamiltonians

We next consider, as defined in Ref. [24], the all-annihilation $H_{aa}(t)$ and all-creation $H_{ac}(t)$ non-Hermitian Hamiltonians, given by

$$H_{aa}(t) = \omega(t)\left(a^\dagger a + 1/2\right) + \alpha(t)a^2, \quad (79a)$$
$$H_{ac}(t) = \omega(t)\left(a^\dagger a + 1/2\right) + \beta(t)a^{\dagger 2}. \quad (79b)$$

These peculiar Hamiltonians, which are extreme forms of the unbalanced Hamiltonian (60), follow when considering $\beta(t) = 0$ and $\alpha(t) = 0$, respectively. To compute their symmetry operators we follow exactly the same procedure presented in subsection 7.1. The same structure of the symmetry operator in Eq. (47) thus applies for these Hamiltonians. Considering a unitary $\Lambda(t)$, the Eqs. (68) simplifies, for the Hamiltonian $H_{aa}(t)$, to

$$\dot{\phi}_{aa} = \omega^*(t) - \omega(-t) + 4\alpha^*(t)\xi^*_{aa}(t)e^{2i\phi_{aa}(t)}, \quad (80a)$$

$$\xi_{aa}(t) = \frac{\alpha(-t) - \alpha^*(t)e^{2i\phi(t)}}{4i \operatorname{Im}\omega(-t)}, \quad (80b)$$

whereas for $H_{ac}(t)$, we obtain

$$\dot{\phi}_{ac} = \omega^*(t) - \omega(-t) + 4\beta^*(t)\xi_{ac}(t)e^{-2i\phi_{ac}(t)}, \quad (81a)$$

$$\xi_{ac}(t) = \frac{\beta(t)e^{2i\phi_{ac}(t)} - \beta^*(-t)}{4i \operatorname{Im}\omega(-t)}. \quad (81b)$$

The nature of the Hamiltonian —following from the group algebra describing the amplification process— is then what defines its symmetry operator. When considering the all-creation and the all-annihilation linear Hamiltonians, derived from the TD linear Hamiltonian introduced above, it can be directly verified that we also have the same structure $I(t) = \mathcal{D}(t)\mathcal{R}(t)\mathcal{T}$ for the symmetry operator.

Therefore, for the Heisenberg algebra of the linear amplification we have the symmetry operator $I(t) = \mathcal{D}(t)\mathcal{R}(t)\mathcal{T}$, whereas for the SU(1,1) algebra of the parametric amplification, we have $I(t) = \mathcal{S}(t)\mathcal{R}(t)\mathcal{T}$. Both symmetry operators share the time reversal and rotation operators. However, as it is well-known from the amplification processes of the radiation field [48,49], while the Heisenberg algebra generates displacements in phase space, the SU(1,1) produces squeezing.

### 7.4 From $I(t)$ in Eq. (47) to $\mathcal{PT}$

As done for the case of a TD non-Hermitian linear Hamiltonian, when considering the constraints under which the Hamiltonian (60) is $\mathcal{PT}$-symmetric [$\omega^*(-t) = \omega(t)$, $\alpha^*(-t) = \alpha(t)$, and $\beta^*(-t) = \beta(t)$], the TI $\mathcal{PT}$ operator can be directly recovered from $I(t)$ in Eq. (47). Under these constraints and considering a unitary $\Lambda(t)$, the Eqs. (68) simplify to

$$
\dot{\phi}(t) = \frac{i}{\operatorname{Im}\omega(-t)}\left\{\alpha^*(t)[\alpha(t)-\beta^*(t)]\left(e^{2i\phi(t)}-1\right)\right.
$$
$$
\left.-\beta^*(t)[\alpha^*(t)-\beta(t)]\left(e^{-2i\phi(t)}-1\right)\right\}, \tag{82a}
$$
$$
\xi(t) = \frac{\alpha^*(t)-\beta(t)}{4i\operatorname{Im}\omega(-t)}\left(1-e^{2i\phi(t)}\right). \tag{82b}
$$

Assuming a TI $\phi = \phi_0 = (2n+1)\pi$, with $n \in \mathbb{Z}$, the Eq. (82a) is automatically satisfied and the Eq. (82b) leads to $\xi = 0$, thus avoiding undesirable constraints on the Hamiltonian parameters. The operator in Eq. (47) is then reduced to

$$
I = e^{-i(2n+1)\pi a^\dagger a}\mathcal{T} = \mathcal{PT}. \tag{83}
$$

## 8 The TI non-Hermitian Hamiltonian of a cavity field under parametric amplification

Considering the TI non-Hermitian Hamiltonian

$$
H = \omega\left(a^\dagger a + 1/2\right) + \alpha a^2 + \beta a^{\dagger 2}, \tag{84}
$$

with complex TD parameters $\omega$, $\alpha$, and $\beta$, we require that $H^\dagger \neq H$, such that $\omega^* \neq \omega$ and/or $\alpha^* \neq \beta$. The $\mathcal{PT}$-symmetry requirement for this TI Hamiltonian imposes the more restrictive conditions $\omega^* = \omega$, $\alpha^* = \alpha$, and $\beta^* = \beta$.

### 8.1 The TI antilinear symmetry operator for the TI Hamiltonian

After computing the TD symmetry in Eq. (47), we assume that its TI equivalent has form $I = \mathcal{S}\mathcal{R}\mathcal{T}$, with $\mathcal{S}$ being an antilinear operator. The condition for the Hamiltonian (84) to be

invariant under this TI symmetry operator, given by $[I, H] = 0$, provides us with the system

$$\omega - \omega^* \cos\left(4\sqrt{\xi\zeta}\right) + \left[\alpha^*\sqrt{\xi^{-1}\zeta}e^{2i\phi} - \beta^*\sqrt{\xi\zeta^{-1}}e^{-2i\phi}\right]\sin\left(4\sqrt{\xi\zeta}\right) = 0, \quad (85\text{a})$$

$$\omega^* - \left[\alpha - \alpha^* e^{2i\phi(t)}\right]\sqrt{\zeta\xi^{-1}}\cot\left(2\sqrt{\xi\zeta}\right) - \left[\alpha - \beta^*\xi\zeta^{-1}e^{-2i\phi}\right]\sqrt{\zeta\xi^{-1}}\tan\left(2\sqrt{\xi\zeta}\right) = 0, \quad (85\text{b})$$

$$\omega^* + \left[\beta - \beta^* e^{-2i\phi}\right]\sqrt{\xi\zeta^{-1}}\cot\left(2\sqrt{\xi\zeta}\right) + \left[\beta - \alpha^* e^{2i\phi}\xi^{-1}\zeta e^{2i\phi}\right]\sqrt{\xi\zeta^{-1}}\tan\left(2\sqrt{\xi\zeta}\right) = 0, \quad (85\text{c})$$

which, for an antiunitary $\mathcal{S}$, i.e., $\zeta = -\xi^*$, simplifies to

$$\omega - \omega^* \cosh\left(4|\xi|\right) + i\left[\alpha^* e^{i\left(2\phi-\varphi_\xi\right)} - \beta^* e^{-i\left(2\phi-\varphi_\xi\right)}\right]\sinh\left(4|\xi|\right) = 0, \quad (86\text{a})$$

$$\omega - \left[\alpha^* - \alpha e^{-2i\phi}\right]e^{i\varphi_\xi}\coth\left(2|\xi|\right) + \left[\alpha^* + \beta e^{-2i\left(\phi+\varphi_\xi\right)}\right]e^{i\varphi_\xi}\tanh\left(2|\xi|\right) = 0, \quad (86\text{b})$$

$$\omega + \left[\beta^* - \beta e^{2i\phi}\right]e^{-i\varphi_\xi}\coth\left(2|\xi|\right) - \left[\beta^* + \alpha e^{-2i\left(\phi-\varphi_\xi\right)}\right]e^{-i\varphi_\xi}\tanh\left(2|\xi|\right) = 0. \quad (86\text{c})$$

Considering a real frequency $\omega$ and, as in the TD case, a small squeezing parameter, $|\xi| \ll 1$, to disregard its second order corrections, we obtain from the above equations the squeezing parameter

$$\xi = \frac{(\alpha - \beta^*) - (\alpha^* - \beta)e^{2i\phi}}{4\omega}, \quad (87)$$

and the rotation angle

$$\phi = \frac{i}{2}\ln\frac{\alpha^* + \beta}{\alpha + \beta^*}, \quad (88)$$

under the constraint $|\alpha|^2 + |\beta|^2 + 2\alpha\beta = 0$. From the polar forms $\alpha = |\alpha|e^{i\varphi_\alpha}$, $\beta = |\beta|e^{i\varphi_\beta}$ and $\xi = |\xi|e^{i\varphi_\xi}$, it follows that

$$|\xi| = \frac{|\alpha|\sin\left(\phi - \varphi_\alpha\right) - |\beta|\sin\left(\phi + \varphi_\beta\right)}{2\omega}, \quad (89\text{a})$$

$$\varphi_\xi = \left(n + \frac{1}{2}\right)\pi - \phi. \quad (89\text{b})$$

For the TI non-Hermitian Hamiltonian (84) we then have the TI antiunitary symmetry operator

$$I = e^{\xi a^2 - \xi^* a^{\dagger 2}}e^{-i\phi a^\dagger a}\mathcal{T}, \quad (90)$$

defined by the parameters in Eqs. (87) and (88). For the case of a $\mathcal{PT}$-symmetric Hamiltonian, where $\alpha$ and $\beta$ are real parameters, the rotation angle following from Eq. (88) is given by $\phi = (2n+1)\pi$, which makes the squeezing degree null, $|\xi| = 0$, thus reducing the symmetry $I = \mathcal{SRT}$ to the $\mathcal{PT}$ operator.

### 8.1.1 Bender-Berry-Mandilara

The validity of the relation $A^{2k} = 1$, with $k$ odd [29], for the symmetry operator in Eq. (90) demands a little algebra. In fact, for the operator in Eq. (90) we obtain

$$\begin{aligned}I^2 &= e^{\xi a^2 - \xi^* a^{\dagger 2}}e^{-i\phi a^\dagger a}e^{\xi^* a^2 - \xi a^{\dagger 2}}e^{i\phi a^\dagger a}\mathcal{T}^2 \\ &= e^{\xi a^2 - \xi^* a^{\dagger 2}}e^{\xi^* e^{2i\phi}a^2 - \xi e^{-2i\phi}a^{\dagger 2}} = 1,\end{aligned} \quad (91)$$

since it follows from Eqs. (89) that $\xi e^{-2i\phi} = -\xi^*$.

## 8.2 Dyson map and pseudo-Hermiticity

For the TI non-Hermitian Hamiltonian (84), we consider the TI Dyson map

$$\eta = \exp\left[\epsilon a^\dagger a + \gamma a^2 + \gamma^* \left(a^\dagger\right)^2\right], \tag{92}$$

again with $\theta^2 = \epsilon^2 - 4|\gamma|^2 > 0$. From the transformation relations in Eq. (72) we obtain

$$h = \eta H \eta^{-1} = W\left(a^\dagger a + \frac{1}{2}\right) + U a^2 + V a^{\dagger 2}, \tag{93}$$

with

$$W = \omega \cosh^2\theta - \left[\omega\left(\epsilon^2 + 4\gamma\lambda\right) - 4(\alpha\lambda + \beta\gamma)\epsilon\right]\frac{\sinh^2\theta}{\theta^2} - 2(\alpha\lambda - \beta\gamma)\frac{\sinh 2\theta}{\theta}, \tag{94a}$$

$$U = \alpha \cosh^2\theta + \left[\alpha\epsilon^2 - 2(\omega\epsilon - 2\beta\gamma)\gamma\right]\frac{\sinh^2\theta}{\theta^2} + (\omega\gamma - \alpha\epsilon)\frac{\sinh 2\theta}{\theta}, \tag{94b}$$

$$V = \beta \cosh^2\theta + \left[\beta\epsilon^2 - 2(\omega\epsilon - 2\alpha\lambda)\lambda\right]\frac{\sinh^2\theta}{\theta^2} - (\omega\lambda - \epsilon\beta)\frac{\sinh 2\theta}{\theta}. \tag{94c}$$

The Hermiticity condition $h = h^\dagger$ demands a real $W$ and $U = V^*$, the former leading to the relation

$$\text{Im}\left[\frac{\epsilon\tanh\theta}{\theta}(\alpha^* - \beta)z - (\alpha^* + \beta)z\right] = 0, \tag{95}$$

where we have defined $z = 2\gamma/\epsilon = z_0 e^{i\varphi}$, with $\gamma = |\gamma| e^{i\varphi}$, such that $-1 \leq z_0 \leq 1$. Considering $\alpha = |\alpha| e^{i\varphi_\alpha}$ and $\beta = |\beta| e^{i\varphi_\beta}$, a particular solution for Eq. (95) follows from the choice of a real $\omega$ and $\varphi = \varphi_\alpha = n\pi - \varphi_\beta$ with an integer $n$, i.e.

$$\omega \in \mathbb{R}, \quad \alpha = |\alpha| e^{i\varphi}, \quad \beta = |\beta| e^{i(n\pi - \varphi)}. \tag{96}$$

The condition $U = V^*$ leads to the relation

$$\epsilon = \frac{1}{2\sqrt{1 - z_0^2}}\tanh^{-1}\left[\frac{\left(|\alpha| - |\beta| e^{-in\pi}\right)\sqrt{1 - z_0^2}}{|\alpha| + |\beta| e^{-in\pi} - \omega z_0}\right], \tag{97}$$

which imposes the restriction $z_- \leq z_0 \leq z_+$, in place of $-1 \leq z_0 \leq 1$, where

$$z_\pm = \frac{\left(|\alpha| + |\beta| e^{-in\pi}\right)\omega \pm \left(|\alpha| - |\beta| e^{-in\pi}\right)\Omega}{\omega^2 + \left(|\alpha| - |\beta| e^{-in\pi}\right)^2}. \tag{98}$$

the frequency $\Omega$ being defined as $\Omega^2 = \omega^2 - 4|\alpha\beta| e^{-in\pi}$, which must be a real parameter as well as $z_\pm$. Although we have a real $\Omega$ for an even $n$, for an odd $n$ we must impose the additional constraint $\omega \geq 2\sqrt{|\alpha\beta|}$ on the Hamiltonian's parameters. For $\omega < 2\sqrt{|\alpha\beta|}$, the pseudo-Hermiticity relation does not apply to the transformed $h$.

Considering the pseudo-Hermiticity conditions in Eq. (96), we obtain from Eq. (89a) the squeezing degree

$$|\xi| = \frac{|\alpha| + (-1)^{n+1}|\beta|}{2\omega}\sin(\phi - \varphi_\alpha), \tag{99}$$

showing that for $\phi = \varphi_\alpha$, the symmetry operator reduces to the product $I = \mathcal{RT}$, whereas for $\phi \neq \varphi_\alpha$ we still have $I = \mathcal{SRT}$, with the squeezing operator being defined by the squeezing degree in Eq. (99) and direction

$$\varphi_\xi = \left(m + \frac{1}{2}\right)\pi - \left(\varphi_\alpha + \frac{1}{2i}\ln\left|\frac{\alpha}{\beta}\right|\right), \quad m \in \mathbb{Z}. \tag{100}$$

### 8.3 Symmetry breaking

Considering the bosonic operators $b$ and $b^\dagger$ defined by the transformations $b = ua + va^\dagger$ and $b^\dagger = v^*a + u^*a^\dagger$, with $|u|^2 - |v|^2 = 1$ such that $\left[b, b^\dagger\right] = 1$, we can rewrite the Hamiltonian (84) in the form

$$H = \Omega\left(b^\dagger b + \frac{1}{2}\right), \tag{101}$$

as long as we consider the constraint $\Omega = \omega/\left[|v|^2 + |u|^2\right] = \alpha/uv^* = \beta/u^*v$, leading to the already defined frequency

$$\Omega = \sqrt{\omega^2 - 4\alpha\beta}. \tag{102}$$

To ensure the reality of the spectrum of the transformed $H$ with a real frequency $\Omega$, we derive from Eq. (102) the same conditions for the pseudo-Hermiticity of the TI Hamiltonian (84), given by Eq. (96). We stress that the parameters in Eq. (96) can also be derived from the symmetry condition $[I, H] = 0$ for a TI Hamiltonian, using the operator in Eq. (90), as done above.

There are two different regimes of eigenvalues following from the parameters given by Eq. (96). One for an odd integer $n$, leading to a real frequency $\Omega$, and the other for an even $n$, where $\Omega$ is real only under the additional constraint $\omega \geq 2\sqrt{|\alpha\beta|}$. For $\omega < 2\sqrt{|\alpha\beta|}$ the symmetry is broken; the frequency is no longer real and the eigenvalues become complex conjugate pairs.

## 9 A connection between the symmetry and metric operators

If the symmetry operator stems from the group algebra of the Hamiltonian, the ansatz for the Dyson's map must also be guided by this group algebra, as discussed in Ref. [12] where a strategy for constructing the Dyson map is presented. It is then reasonable to assume that there must be a close connection between both of these operators: the symmetry and the Dyson map (or equivalently the metric operator).

In part I of this work we have derived the symmetry operator for the TD non-Hermitian linear Hamiltonian, which resulted in the form $I(t) = \mathcal{D}(t)\mathcal{R}(t)\mathcal{T}$, with $\mathcal{R}(t) = e^{-i\phi(t)a^\dagger a}$ being a unitary operator and $\mathcal{D}(t) = e^{v(t)a^\dagger + \lambda(t)a + \mu(t)}$ a unitary ($\lambda = -v^*$) or a Hermitian nonunitary operator ($\lambda = v^*$). The ansatz for the Dyson map we have used for the pseudo-Hermitization of this linear Hamiltonian, given by the Hermitian operator $\eta(t) = e^{\epsilon(t)a^\dagger a + \gamma(t)a + \gamma^*(t)a^\dagger}$, bears resemblance to the generalized displacement $\mathcal{D}(t)$. In fact, we could have considered for the Dyson map the nonunitary operator $\mathcal{D}(t)\mathcal{R}(t)$ instead of the $\eta(t)$, possibly at the expense of constraints on the non-Hermitian Hamiltonian's parameters. We could also have associated the linear part of the symmetry directly to the metric operator, i.e., $\rho(t) = \mathcal{D}(t)\mathcal{R}(t)$.

In the present work, we have obtained the symmetry operator $I(t) = \mathcal{S}(t)\mathcal{R}(t)\mathcal{T}$, for the TD non-Hermitian quadratic Hamiltonian, with $\Lambda(t) = e^{\xi(t)a^2 + \zeta(t)a^{\dagger 2}}$, being a unitary ($\zeta = -\xi^*$) or nonunitary Hermitian operator ($\zeta = \xi^*$). The Dyson map we have used for the pseudo-Hermitization of our quadratic Hamiltonian, $\eta(t) = \exp[\epsilon(t)a^\dagger a + \gamma(t)a^2 + \gamma^*(t)\left(a^\dagger\right)^2]$, also resembles the nonunitary $\mathcal{S}(t)\mathcal{R}(t)$, which could be used for the Dyson map instead of $\eta(t)$. Alternatively, as anticipated above, we could also have proposed the relation $\rho(t) = \mathcal{S}(t)\mathcal{R}(t)$.

Considering the TD non-Hermitian quadratic Hamiltonian and assuming $\eta(t) = \mathcal{S}(t)\mathcal{R}(t)$, the above discussion leads us to propose the relation

$$I(t) = \eta(t)\mathcal{T}. \tag{103}$$

With the Dyson map

$$\eta(t) = \mathcal{S}(t)\mathcal{R}(t), \tag{104}$$

we obtain from Eq. (60) the transformed Hamiltonian

$$h = \tilde{W}\left(a^\dagger a + \frac{1}{2}\right) + \tilde{U}a^2 + \tilde{V}a^{\dagger 2}, \tag{105}$$

with

$$\tilde{W} = \left(\dot{\phi} + \omega\right)\cos\left(4|\nu|\right) - \left[\alpha e^{-i(2\phi - \varphi_\nu)} - \beta e^{+i(2\phi - \varphi_\nu)}\right]\sin\left(4|\nu|\right), \tag{106a}$$

$$\tilde{U} = \frac{\dot{\phi} + \omega}{2}\sin\left(4|\nu|\right)e^{-i\varphi_\nu(t)} + i\dot{\nu}^* + \alpha e^{-2i\phi}\cos^2\left(2|\nu|\right) + \beta e^{2i(\phi - \varphi_\nu)}\sin^2\left(2|\nu|\right), \tag{106b}$$

$$\tilde{V} = -\frac{\dot{\phi} + \omega}{2}\sin\left(4|\nu|\right)e^{i\varphi_\nu(t)} + i\dot{\nu} + \alpha e^{-2i(\phi - \varphi_\nu)}\sin^2\left(2|\nu|\right) + \beta e^{2i\phi}\cos^2\left(2|\nu|\right). \tag{106c}$$

For the Hamiltonian (105) to be Hermitian, it follows the system

$$0 = |\omega|\sin\varphi_\omega + \left[+|\alpha|\sin\left(2\phi - \varphi_\nu - \varphi_\alpha\right) + |\beta|\sin\left(2\phi - \varphi_\nu + \varphi_\beta\right)\right]\tan\left(4|\nu|\right), \tag{107a}$$

$$\dot{\varphi}_\nu = -\frac{\cos\left(4|\nu|\right)}{2|\nu|}\left[\left(\dot{\phi} + |\omega|\cos\varphi_\omega\right)\tan\left(4|\nu(t)|\right) + |\alpha|\cos\left(2\phi - \varphi_\nu - \varphi_\alpha\right) \right.$$
$$\left. - |\beta|\cos\left(2\phi - \varphi_\nu + \varphi_\beta\right)\right], \tag{107b}$$

$$|\dot{\nu}| = \frac{1}{2}\left[|\alpha|\sin\left(2\phi - \varphi_\nu - \varphi_\alpha\right) - |\beta|\sin\left(2\phi - \varphi_\nu + \varphi_\beta\right)\right], \tag{107c}$$

which, like that in Eq. (76), does not lead to constraints on the parameters of Hamiltonian (60). Therefore, the Dyson map (104) can perfectly be used as an alternative to the ansatz in Eq. (71), thus leading to the associated metric operator $\rho(t) = \mathcal{R}^\dagger(t)\mathcal{S}^\dagger(t)\mathcal{S}(t)\mathcal{R}(t)$.

When we consider Eq. (103) together with the relation $I(t) = \Xi^{-1}(t)\rho(t)$ derived above, we obtain $\eta(t) = \Xi^{-1}(t)\rho(t)\mathcal{T}$, and consequently

$$\Xi(t) = \rho(t)\mathcal{T}\eta^{-1}(t). \tag{108}$$

When we consider, however, the ansatz

$$I(t) = \mathcal{T}\eta(t), \tag{109}$$

as an alternative to Eq. (103), we get a simplest form for the $\Xi(t)$-anti-pseudo-Hermitian operator, given by

$$\Xi(t) = \eta^\dagger(t)\mathcal{T}. \tag{110}$$

Therefore, after computing the TD symmetry operator through Eq. (9), we automatically derive the Dyson map from the symmetry-metric Eq. (103), and consequently the metric operator or the $\rho$-pseudo-Hermitian operator. Then, from Eq. (110), we derive the $\Xi$-anti-pseudo-Hermitian operators. In short, after computing the symmetry operator, all the other operators thus follow automatically.

We observe however that, from Eqs. (9) and (103), we directly derive an equation for the Dyson map, given by

$$i\frac{\partial \eta(t)}{\partial t} = \eta(t)\mathcal{T}H(t)\mathcal{T}^{-1} - H(-t)\eta(t). \tag{111}$$

From Eqs. (109) and (103), we obtain instead:

$$i\frac{\partial \eta(t)}{\partial t} = \mathcal{T}H(-t)\mathcal{T}^{-1}\eta(t) - \eta(t)H(t). \tag{112}$$

Eqs. (111) or (112) can be solved by applying the same reasoning used in Section 3 to compute the symmetry operator.

We finally note that the formal simplicity of $\mathcal{PT}$ symmetry prevented us, so far, from exploring a relation between symmetry and metric as anticipated here. It is evident that our proposal for this relation was largely due to the complexity of the symmetry operators associated with the Hamiltonians discussed here. This fact should motivate us to persist in the investigation of symmetries that are more general than $\mathcal{PT}$, through the consideration of other Hamiltonians with different symmetry groups. We believe that we can further deepen our understanding of pseudo-Hermitian systems and even the symmetry-metric relation through this investigation.

# 10 Conclusions

In this work we have proposed a method for the derivation of general TD continuous symmetry operators for TD non-Hermitian Hamiltonians. Although our method applies indistinctly to linear or antilinear, unitary or nonunitary symmetries, we then assume an antilinear symmetry to retrieve the results by Mostafazadeh [4] and Bender-Berry-Mandilara [29] for the case of TI Hamiltonian and symmetry operators. In fact, assuming that the TD non-Hermitian Hamiltonian is simultaneously $\rho$-pseudo-Hermitian and $\Xi$-anti-pseudo-Hermitian, we derive the relation $I(t) = \Xi^{-1}(t)\rho(t)$ for our TD antilinear symmetry operator. From this relation we recover the Mostafazadeh's theorem, for TI Hamiltonian and symmetry operators, asserting that the pseudo-Hermiticity of a Hamiltonian implies the existence of an antilinear symmetry of the form $I = \Xi^{-1}\rho$. We also retrieve the Bender-Berry-Mandilara result that a non-Hermitian Hamiltonian presents a real spectrum when invariant under any antiunitary operator $I$ satisfying $I^{2k} = 1$ with $k$ odd.

Our method is also based on a proposal in Ref. [44], for the construction of Lewis & Riesenfeld TD nonlinear invariants, and we have applied it for the case of TD non-Hermitian linear and quadratic Hamiltonians modelling a cavity field under linear and parametric amplifications. We have thus derived TD continuous symmetry operators, given in Eqs. (30) and (70), for the linear and quadratic cases respectively. These operators describe the successive actions of a time-reversal operator $\mathcal{T}$, a TD rotation and a generalized displacement or squeezing in phase space, respectively. This TD continuous symmetry automatically reduces to the TI discrete $\mathcal{PT}$ operator when we restrict our TD Hamiltonian to be $\mathcal{PT}$-symmetrical.

After computing the symmetry operators we then consider the pseudo-Hermitization of our TD linear and quadratic Hamiltonians. For the case of the TI equivalents of our non-Hermitian Hamiltonians, the stringent invariance requirement $[I, H] = 0$, imposes TI continuous symmetry operators which are very particular case of the TD symmetry operators in Eq. (30) and (70), even though they are generalizations of the discrete parity and time-reversal transformation. The TI non-Hermitian Hamiltonians and the associated symmetry operators are more vulnerable, as expected, than their general TD equivalents to the constraints imposed by the pseudo-Hermiticity relation.

The TD general symmetries in Eq. (30) strongly resemble the evolution operator for the Hermitized counterparts of our TD non-Hermitian linear and quadratic Hamiltonians, except for the time-reversal operation [25–27,45,46]. If applied to a given state of these Hermitized counterparts of our Hamiltonians, these peculiar symmetry operators cause the probability distribution to trace an upward spiral in phase space, with TD rotation and translation rates, for the linear case, or translation plus squeezing rates for the quadratic case.

We have also computed the TD symmetry operators for the all-creation and all-annihilation Hamiltonians, in Eqs. (79a) and (79b), which are extreme cases of the parametric amplification in Eq. (60), where only creation or annihilation quadratic operators take place in the

interaction picture. We verify, as expected, that the TD symmetry operators for these peculiar Hamiltonians have the same form as the symmetry operator for our non-Hermitian parametric amplification. The group algebra describing the Hamiltonian is what defines its symmetry operator.

From the results we have derived for the linear and the quadratic non-Hermitian Hamiltonians, we have proposed a relation between the symmetry and the metric operators which enables us to automatically derive the metric operator $\rho(t)$. In fact, from the symmetry of the problem, we are able to derive both the $\rho$-pseudo-Hermitian and the $\Xi$-anti-pseudo-Hermitian operators, as introduced by Mostafazadeh in Ref. [4]. In order to consolidate the symmetry-metric relation which we have derived, our next step is to approach other Hamiltonians, associated with different groups of symmetries.

Therefore, in the present work we have explored more general symmetries than parity and time-reversal for TD non-Hermitian Hamiltonians. The TD continuous symmetry operators we have derived follow evidently from the symmetry group of the Hamiltonians we have studied, and the symmetry-metric relation we have proposed will certainly act as a guide for the study of pseudo-Hermitian quantum mechanics for symmetry groups far broader than $\mathcal{PT}$.

# Acknowledgements

The authors would like to thank CAPES, CNPq, and INCT-IQ, for support.

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
