# Peer review of "Beyond PT-symmetry: Towards a symmetry-metric relation for time-dependent non-Hermitian Hamiltonians"

_SciPost Physics, doi:SciPost Phys. Core 5, 012 (2022)_

## Round 1 · Referee Report · Anonymous · 2021-11-12

Strengths

This is the first attempt to extend two basic theorems on time-independent non-Hermitian operators to time-dependent Hamiltonian operators.

Weaknesses

Lacks physical motivation. Its presentation may be substantially improved. The paper can made much shorter.

Report

The current version does not meet the standards of the journal. But I think the authors can be given a chance to respond to the criticisms I have listed in my report. An improved version may qualify for publication in SciencePost and other more specialized journals.

Requested changes

See the attached PDF.

Attachment

  • validity: ok
  • significance: ok
  • originality: good
  • clarity: ok
  • formatting: acceptable
  • grammar: below threshold

---

## Round 2 · Referee Report · Anonymous (Referee 1) · 2022-1-16

Strengths

It provides time-dependent generalizations of certain basic results on time-independent pseudo-Hermitian Hamiltonians.

Weaknesses

  • It lacks a discussion of the physical aspects of the results.
  • It involves a number of formal assumptions with no physical justification.
  • The lengthy manipulations performed to deal with the specific toy models considered in the paper do not lead to any physical results.

Report

The authors have improved the presentation of their paper substantially and addressed most of the points I had raised in my initial report (or instead provided some explanation about them in their response letter). However, my main objection is still valid; I do not see how the developments reported in this paper help solve or shed light on an outstanding physics problem. As far as I can see, the authors' findings do not lead to a specific prediction of a new physical effect nor do they relate to a known physical effect. I am also unable to think of a potential physical application for them. Therefore, I cannot qualify them as “groundbreaking results.” They are theoretical developments that should be of interest for experts working on the formal aspects of non-Hermitian Hamiltonians. I suggest the authors to seek publication of their work in a more specialized theoretical/mathematical physics journal.
  • validity: high
  • significance: ok
  • originality: ok
  • clarity: ok
  • formatting: acceptable
  • grammar: good

Author:  Miled Moussa  on 2022-01-19  [id 2108]

(in reply to Report 1 on 2022-01-16)

To The SciPost Physics Team.

Thank you for your message of 16 Jan, regarding our submitted manuscript scipost-202110-00013v1. We also thank the referee efforts. However, we vehemently disagree with the referee's views, and below we make our reasons clear:

1) In the report (on 2022-1-16) the referee states that:

"The authors have improved the presentation of their paper substantially and addressed most of the points I had raised in my initial report (or instead provided some explanation about them in their response letter). However, my main objection is still valid; I do not see how the developments reported in this paper help solve or shed light on an outstanding physics problem. As far as I can see, the authors' findings do not lead to a specific prediction of a new physical effect nor do they relate to a known physical effect. I am also unable to think of a potential physical application for them. Therefore, I cannot qualify them as "groundbreaking results." They are theoretical developments that should be of interest for experts working on the formal aspects of non-Hermitian Hamiltonians. I suggest the authors to seek publication of their work in a more specialized theoretical/mathematical physics journal."

Indeed we have not presented a "new physical effect", but we have taken a significant step in the construction of the quantum mechanics of time-dependent (TD) pseudo-Hermitian Hamiltonians and metric operators. More specifically, with the aim of exploring more general symmetries than PT in the general scenario of TD non-Hermitian Hamiltonians, we have provided, for the first time in the literature, a method for the derivation of TD symmetries associated with these TD Hamiltonians which applies indistinctly to linear or antilinear, unitary or nonunitary symmetries. As a way of proving the validity of our developments, we have then assumed the symmetry to be an antilinear operator to retrieve the results by Mostafazadeh [3] and Bender-Berry-Mandilara [22] in the particular case of a TI scenario, i.e., TI non-Hermitian Hamiltonians, metrics and symmetries. Indeed, the above mentioned theorem by Mostafazadeh [3] is retrieved when considering an antilinear symmetry while the result by Bender-Berry-Mandilara is retrieved when considering a unitary antilinear or antiunitary symmetry. In the case where the TD symmetry is assumed to be a linear (instead of antilinear) transformation, we obtain the equation which defines a dynamical invariant for a non-Hermitian Hamiltonian, showing, again for the first time in the literature, that a TD (linear) symmetry operator is the dynamical invariant for a non-Hermitian Hamiltonian. (And here we note that we must make a small change in the manuscript, modifying the sentence below Eq. (10) to its correct form: "which defines a linear dynamical invariant [29-31] for a non-Hermitian Hamiltonian H(t).) In addition, guided by the results in Refs. [2,22], we have also explored the connection between antilinear symmetries and metrics. We have derived a relation between the TD symmetry and a pair of TD metric operators [I(t)=Ξ⁻¹(t)ρ(t)], one linear [ρ(t)] and the other antilinear [Ξ(t)], which is analogous to the Mostafazadehís relation [3] (I =Ξ⁻¹ρ) for the TI scenario. This connection between symmetry and metric is explored a little further, leading us to propose a relation between symmetry (I) and metric (ρ) operators [I(t) = eta(t)T ]. Then, this symmetry-metric relation is put in perspective with the TD antilinear symmetry I(t) = Ξ⁻¹(t)ρ(t) we have derived, allowing us to finally compute the Ξ(t)-anti-pseudo-Hermitian operator. In short, following the path opened by Mostafazadeh in 2002, and in an attempt to adapt his formalism to the scenario of TD non-Hermitian Hamiltonians and metric operators, we have thus presented solid advances that can contribute to the debate of TD pseudo-Hermitian quantum mechanics, which we believe, do make our manuscripts suitable for publication in SciPost Physics.

Regarding the referee's claim that "I do not see how the developments reported in this paper help solve or shed light on an outstanding physics problem", we reply that when treating a complex physical problem involving a TD non-Hermitian Hamiltonian, we now know from the developments presented in our manuscript, that it is possible to compute a general TD symmetry I(t) from equation idI(t)/dt + H(t)I(t) I(t)H(t) = 0. Then we automatically compute the Dyson map eta(t) using the symmetry-metric relation I(t) = eta(t)T , and consequently the ?ρ(t)-pseudo-Hermitian metric operator and the Ξ(t)-anti-pseudo-Hermitian operator Ξ(t) = eta_dagger(t)T . All these proposed equations were tested with two examples presented in detail: the processes of linear and parametric amplifications of a cavity mod e.

The referee then claims that "As far as I can see, the authors' findings do not lead to a specific prediction of a new physical effect nor do they relate to a known physical effect. I am also unable to think of a potential physical application for them". Indeed, we are not looking for a new physical effect in our manuscript, we are exploring the quantum mechanics of TD pseudo-Hermitian Hamiltonians beyond PT symmetries. Regarding the potential physical application for them, we reinforce, as we did in our previous answer to the referee, that with the advances in radiation-matter interaction platforms, we have reasons to believe that we will soon be engineering time-dependent processes of linear and parametric amplification, as those described in our manuscripts, or even time-dependent Josephson-type coupling in two-mode Bose-Einstein condensates and the consequences of this time-dependent coupling for the associated phases transitions. Moreover, non-Hermitian Hamiltonians are becoming a topic of great interest in all areas of physics. We mention, for example, the works on Majorana bound states and pairing in non-Hermitian superconducting, both driven by non Hermiticity.

Next, we address the referee's comment that the results in our manuscript "are theoretical developments that should be of interest for experts working on the formal aspects of non-Hermitian Hamiltonians. I suggest the authors to seek publication of their work in a more specialized theoretical/mathematical physics journal." In view of all our new results mentioned above (a method for the derivation of the general TD symmetry, the finding that the TD linear symmetry is a dynamic invariant associated with the non-Hermitian Hamiltonian, the generalization of the results of Mostafazadeh [3] and Bender-Berry-Mandilara [22], and the new equation relating symmetry and metric), we have no doubt that our manuscript is perfectly suitable for SciPost Physics.

We would also like to comment on the points that the referee indicated as weaknesses in our paper: 1) It lacks discussion on the physical aspects of the results. 2) It involves a number of formal assumptions with no physical justification. 3) The lenghtly manipulation performed to deal with the specific toy models considered in the paper do not lead to any physical results.

Regarding the first point, we do have presented a lenghtly discussion of the physical results. For example, all the physical discussion presented in this letter is entirely contained in the manuscript. Regarding the point 2), the only formal assumption we have made concerns the symmetry-metric relation I(t) =eta(t)T . There is no other formal assumption in the manuscript. Finally, regarding point 3), we first mention that there are no toy models in our manuscript; both of our examples are realistic physical systems deeply studied in the quantum optics literature, both theoretically and experimentally. Then, our lenghtly manipulation performed to deal with the linear and parametric amplification cases, lead to our proposed symmetry-metric relation and all the consequences we have discussed that follows from this relation presented for the first time in the literature.

Finally, it is important to emphasize that in his/her second letter, the referee did not contest our answers that demanded formal knowledge in the field of quantum mechanics of pseudo-Hermitian Hamiltonians. In fact, the referee did not contest our answers to questions 3 to 9, as posed in our first response, showing that the referee was wrong in all these 7 points.

We are completely sure of the importance and correctness of the work we have done, and we cordially request the SciPost team to make a decision based on the results we have presented.

Best regards, L. F. A. da Silva, R. A. Dorado, and M. H. Y. Moussa.

Attachment:

To-The-SciPost-Physics-Team.pdf

---

## Round 2 · Author Response

To The SciPost Team.

Thank you for your message of 21 Nov, regarding our submitted manuscript scipost-202110-00013v1. We also thank the referee for the constructive comments and suggestions. We are now presenting a new version of the manuscript, in which we have merged the two previous versions into one. Furthermore, following the referee's suggestions, in the new version of the manuscript we i) introduced additional paragraphs for a better presentation of our objectives, ii) improved the writing of some sentences, and iii) added missing references in the first version. We respectfully disagree with some of the points raised by the referee and set out in detail the reasons for our disagreements. Below we present all the modifications made to the manuscript and all the answers to the questions raised by the referee. As should be clear from our responses to the referee's questions, substantial changes have been made to the manuscript, making it more concise and clearer.

1) The first point raised by the referee is: The introduction is too wordy. Parts of it seem redundant or unnecessary. There is no discussion of the concrete motivation for this work. Is there a specific physics problem that the authors' results can shed light upon? Or is this an example of generalize because you can?.

This is an important question. Despite all our efforts to make clear the motivation of our work, it seems that we were not successful. In Part I of the previous version, our objectives were basically exposed in the fifth and sixth paragraphs of the Introduction. In the new merged version we substantially improve the presentation of the goals by introducing other paragraphs and changing the order of paragraphs from the previous version. We now have:

"Our objective is precisely to explore more general symmetries than PT starting from the general scenario of TD non-Hermitian Hamiltonians. The method we propose for the derivation of TD symmetries for TD non-Hermitian Hamiltonians applies indistinctly to linear or antilinear, unitary or nonunitary symmetries. However, we assume the symmetry to be an antilinear operator aiming to retrieve the results by Mostafazadeh [3] and Bender-Berry-Mandilara [22] in the particular case of a TI scenario, i.e., TI non-Hermitian Hamiltonians, metrics and symmetries. The above mentioned theorem by Mostafazadeh [3] is retrieved when considering an antilinear symmetry while the result by Bender-Berry-Mandilara is retrieved when considering a unitary antilinear or antiunitary symmetry.
After presenting our method to derive the symmetry operator, we then apply it for TD non-Hermitian linear and quadratic Hamiltonian, modelling a cavity field under linear and parametric amplifications. As expected, we have derived TD continuous antilinear symmetries far more complex than the spatial reflection and time reversal. These TD symmetries are then particularized to the equivalent TI non-Hermitian linear and quadratic Hamiltonians and their PT-symmetric restrictions. Then, in the TI scenario, the results in Refs. [3,22] are perfectly retrieved. The PT-symmetry is then a particular case of more general symmetries in which spatial reflection is generalized to continuous rotations followed by additional displacement and/or squeezing in phase space. Our results reinforces the prospects of going beyond PT-symmetric quantum mechanics making the field of pseudo-Hermitian quantum mechanics even more comprehensive and promising.
In addition, guided by the results in Refs. [2,22], here we explore the connection between antilinear symmetries and metrics. We derive a relation between the TD symmetry and a pair of TD metric operators, one linear and the other antilinear, which is analogous to the Mostafazadeh's relation [3] for the TI scenario. This connection between symmetry and metric is explored a little further, leading us to propose a relation between symmetry (I) and metric (ρ) operators. Then, this symmetry-metric relation is put in perspective with the TD antilinear symmetry I(t)=Ξ⁻¹(t)ρ(t) we have derived, allowing us to finally compute the Ξ(t)-anti-pseudo-Hermitian operator.
We stress that from the 1990s onwards, the field of radiation-matter interaction underwent extraordinary progress when experimentalists began to coherently control the process of the interaction of a single atom with a single photon of the radiation field [25]. This control allowed probing fundamental aspects of quantum mechanics and implementing quantum logic operations, among other important achievements. We also remember the construction of the Bose-Einstein condensates, which allowed unprecedented control in the manipulation of many-body processes [26]. For some time now, this control of the atom-field interaction has been sought towards time-dependent processes of radiation-matter interaction, as we know from the many advances made in the experimental verification of the dynamical Casimir effect [27]. We have reason to believe that not only the dynamical Casimir effect, but other processes involving TD Hamiltonians ---such as a TD Josephson-type coupling in two-mode Bose-Einstein condensates [28]--- will soon be achieved. Therefore, the TD pseudo-Hermitian quantum mechanics, in its most general form, accounting for TD non-Hermitian Hamiltonians, symmetries and metric operators, must be studied to account for these TD processes.
We also note that our results shed light on the treatments already presented in the literature on TD non-Hermitian Hamiltonians. For example, our developments considerably broaden our understanding of those presented in Refs. [10,11,19-21, where all the analysis on symmetry is reduced to the conditions for a TD Hamiltonian to be PT-symmetric. From our conclusions, we now known that there is close connections between the group algebra of the Hamiltonian, the symmetry and the metric operator. As concluded bellow, when considering a TD non-Hermitian and non-PT-symmetric Hamiltonian, we can now start by computing the symmetry operator I(t) of the system modeled by the Hamiltonian H(t), from which we automatically compute the metric operator, the ρ-pseudo-Hermitian and the Ξ-anti-pseudo-Hermitian operators."

With all due respect to the referee's opinion, we are focusing on time-dependent pseudo-Hermitian processes because we need to study them, and not exactly because we can. We allow ourselves to say that it would be worthwhile to study these time-dependent problems even if they were just theoretical abstractions. However, with the advances in radiation-matter interaction platforms, we indeed have reasons to believe that we will soon be engineering time-dependent processes of linear and parametric amplification, as those described in our manuscripts, or time-dependent Josephson-type coupling in two-mode Bose-Einstein condensates and the consequences of this time-dependent coupling for the associated phase transitions, etc...

We finally note that as early as 2007, Mostafazadeh [A. Mostafazadeh, Phys. Lett. B 650, 208 (2007)] raised the issue of time-dependent non-Hermitian Hamiltonians treated through equally time-dependent metric operators, thus inaugurating this important path within the quantum mechanics of pseudo-Hermitian Hamiltonians. In the present manuscript we follow the path opened by Mostafazadeh in 2002 in an attempt to adapt his formalism to the scenario of time-dependent non-Hermitian Hamiltonians and metric operators, and we do present solid advances that can contribute to the debate, which makes our manuscripts suitable for publication in SciPost.

2) The second point raised by the referee is: The authors miss to cite the early publications where time-dependent pseudo-Hermitian operators were originally studied. See form example quant-ph/0306200, quant-ph/0604014, arXiv:0706.1872, and references therein.

In the new version of the manuscript, at the end of the second paragraph, we fixed this problem through the sentences: "The consequences of allowing the Hilbert space to have a TD metric was posed in Ref. [14], while a treatment of a TD non-Hermitian Hamiltonian through a TI metric operator was done in Ref. [15]. The investigation of TD metric operators for the treatment of TD non-Hermitian Hamiltonians was undertaken in Ref. [16]."

The absence of Ref. [16] in our previous versions of the manuscripts was really a flaw, now properly repaired. However, the Ref. [15], although important, does not consider time-dependent metric operators, the subject of our manuscripts. However, it is also quoted, making our introduction more precise.

3) The third point raised by the referee is: The authors claim that "The PT-symmetry condition," is "much less demanding than that of Hermiticity, · · · ." This is simply false, because PT-symmetric Hamiltonians H=p²+v(x), with the standard definition of P and T , satisfy H^{†}=PHP which means that HP is Hermitian. Therefore demanding PT-symmetry of H is equivalent to demanding Hermiticity of HP.

We note that by "much less demanding than that of Hermiticity" we mean, following C. Bender [Contemp. Phys. 46, 277 (2005)] that: "The central idea of PT-symmetric quantum theory is to replace the condition that the Hamiltonian of a quantum theory be Hermitian with the weaker condition that it possess space-time reflection symmetry (PT symmetry). This allows us to construct and study many new kinds of Hamiltonians that would previously have ignored."

We must also stress that we are not sure we have understood this point raised by the referee. In fact, it is clear that even regarding the Hamiltonian considered by the referee, H=p²+v(x), the condition for PT-symmetry, given by v^{∗}(2x₀-x)=v(x) (defining P as a reflection about whatever x₀) is weaker than that for Hermiticity, given by v^{∗}(x)=v(x). For the case of the parametric amplification (we could have considered the linear amplification as well), with H=ωa^{†}a+ α(a^{†})²+βa², the condition for Hermiticity, given by ω∈ℝ and β=α^{∗}, is even stronger that that for PT-symmetry, demanding ω,α,β∈ℝ.

In the new version of the manuscript we have modified a little our claim to: "The PT-symmetry condition, weaker than that for Hermiticity, greatly expands the possibility of the Hamiltonian description of physical systems (with real eigenvalues [1] and conservation of the norm [2]), which is one of the strong calls for the field."

4) The fourth point raised by the referee is: In page 4 the authors write: "This in a way gives us physical support to extend the scope of pseudo-Hermitian Hamiltonians beyond those invariants under PT operation." This gives the impression that the scope of pseudo-Hermitian Hamiltonians was previously limited to PT-symmetry which is not true. Pseudo-Hermiticity was introduced in an attempt to provide a general framework that would encompass PT -symmetry and clarify the multitude of claims about its niceties.

In fact, in this sentence we expressed ourselves poorly. In the new version of the manuscript we have rewritten this sentence to: "This in a way gives us physical support to explore the application of the general framework of pseudo-Hermiticity for non-Hermitian Hamiltonians beyond those invariants under PT operation".

5) The fifth point raised by the referee is: The authors define time-dependent pseudo-Hermiticity via eq. (2). If ρ is time-independent this reduces to "ρ-pseudo-Hermiticity" which is, strictly speaking, not the same as "pseudoHermiticity." The latter demands the existence of an invertible Hermitian operator ρ (which needs not be positive-definite) satisfying eq. (7). Eq. (2) should be identified with the condition for "ρ(t)-pseudo-Hermiticity for a TD Hamiltonian" where ρ(t) is a given positive-definite metric operator. This is actually not an unimportant technicality. If we identify TD pseudo-Hermiticity by the condition of the existence of a metric operator ρ(t) such that (2) holds, we essentially put no restriction on the Hamiltonian H(t). This is because we can solve (2) for ρ(t) and determine ρ(t) for every H(t). This is actually done in math-ph/0209014. For this reason the authors must make it clear that they fix ρ(t) first and study time-dependent ρ(t)-pseudo-Hermitian Hamiltonians which are defined by their eq. (2). Of course this leads one to the question: "How and why should one fix a particular ρ(t) and study the Hamiltonians that satisfy eq. (2)?

The concepts of pseudo-Hermiticity and ρ-pseudo-Hermiticity is very well discussed in Ref. [A. Mostafazadeh, Entropy 22, 471 (2020)], the latter being a particular case of the former. In this reference, Mostafazadeh stress that: "With the stronger requirement that ρ be positive-definite one can establish the reality of the spectrum of H, its quasi-Hermiticity (existence of a positive-definite automorphism ρ=√ρ such that h := ρ⁻¹Hρ is Hermitian), and the exactness of the antilinear symmetry Ξ ." However, as usual in the literature, in our manuscript we only use pseudo-Hermiticity or TD pseudo-Hermiticiy, without differentiating pseudo-Hermiticity from ρ-pseudo-Hermiticity. Just as an example, all this concepts or aspects are discussed in Ref. [H. F. Jones, J. Phys. A 42, 135303 (2009)] without differentiating pseudo-Hermiticity from ρ-pseudo-Hermiticity. H. Jones also remarks that "...it is frequently extremely useful to write (the metric) in the exponential form ρ=√ρ=e^{-Q}", as we did in our manuscript.

On the other hand, we are not sure we have correctly understood the referee's subsequent considerations: If we identify TD pseudo-Hermiticity by the condition of the existence of a metric operator ρ(t) such that (2) holds, we essentially put no restriction on the Hamiltonian H(t).

The possibility of computing the time-dependent (TD) parameters of the metric operator from Eq. (2), without imposing restrictions on the TD parameters of the Hamiltonian, is exactly what we want. On this regard, we must always consider the more general ansatz for the metric operator ---by defining this operator through the symmetry group of the Hamiltonian itself--- so that the determination of the parameters of the metric does not imply restrictions on the parameters of the Hamiltonian. TD metric operators (or Dyson maps) are considered precisely to prevent the occurrence of restrictions on the TD parameters of the Hamiltonians themselves. The use of a TI metric for the treatment of a TD non-Hermitian Hamiltonian will definitely impose restrictions on the Hamiltonian's parameters.

Regarding the referee's claim: For this reason the authors must make it clear that they fix ρ(t) first and study time-dependent ρ(t)-pseudo-Hermitian Hamiltonians which are defined by their eq. (2), we believe that it's clear from the manuscript that we have started from a general ansatz for ρ(t) and then computed its TD parameters such that Eq. (2) holds.

6) The sixth point raised by the referee is: In Section III, the authors write "we assume the symmetry to be TD operator." in which "symmetry" should change to "symmetry generator." I do not understand the reason for this assumption. Indeed calling I(t) a "symmetry" or "symmetry generator" is rather inappropriate. This is because eq. (9) is more of a restriction on I(t) than a restriction on H(t). In practice a quantum system is determined by a given Hamiltonian operator and one solves (9) to find an invariant operator I(t).

Concerning the nature of the TD symmetry operator I(t), we have solid reasons that motivated us to define it:

i) Fist of all, from the equation defining our TD symmetry

i(∂/(∂t))I(-t)|ψ(-t)>=(I(-t)H(-t)I⁻¹(-t)+i((∂I(-t))/(∂t))I⁻¹(-t))I(-t)|ψ(-t)>,  <label>1</label>

with

i((∂I(t))/(∂t))+H(-t)I(t)-I(t)H(t)=0.  <label>2</label>

it follows that the transformation I(t) produces an independent solution I(-t)|ψ(-t)> of the Schrödinger equation from a given solution |ψ(t)>, as demanding by the definition of symmetry [see for example W. Greiner and B. Müller, Quantum Mechanics - Symmetries, Springer-Verlag, New York (2992)]. ii) Moreover, from Eq. (<ref>2</ref>) we consistently retrieve the well established equations for the particular cases of time-independent symmetries and/or Hamiltonians. Therefore, why not define this generalized equation, really simple, but so far absent in the literature? We would like to see this result (along with the others which support it in the manuscripts) published, so that the community as a whole can judge its effectiveness.

iii) In both applications we made, for the linear and parametric amplifications, we found that the symmetry operators obtained through the above equation are, as expected, directly associated with the symmetry groups of the Hamiltonians, and therefore directly associated with the evolution operators as clearly outlined in the manuscript. These time-dependent continuous symmetry operators make perfect sense when we analyze the actions they represent in phase space, shedding light on the dynamics of the physical processes; they anticipate the dynamics of the state vectors that result from the actions of the evolution operators. Thus, they are perfectly in line with our expectations, helping us to deepen the field of application of pseudo-Hermitian quantum mechanics beyond PT-symmetry.

As a consequence of what was said above, we conclude that the I(t), as defined by equation (9), is certainly a restriction on I(t), but a restriction determined by the Hamiltonian H(t), which automatically follows from the symmetry group of H(t) and therefore bears a close relation with the evolution operator that follows from H(t). And this conclusion is perfectly supported by the two applications we made in the manuscript.

iv) This symmetry operator allowed us for the first time in the literature to conjecture an equation that relates symmetry and metrics. We have it perfectly in mind that the criticisms raised by the referee are constructive and help to make the manuscripts more solid and clear; after all, he wants to hear our response to his/her sever criticisms. But again we ask why not let the community judge this equation which for the first time relates symmetry and metric?

7) The seventh point raised by the referee is: I do not think eq. (10) defines a dynamical invariant when H(t) is a non-Hermitian Hamiltonian operator. This is actually easy to see. One can write anti-linear invariant of eq. (9) as TI(t) where I(t) is linear, and use H(-t)T=TH^{†}(t) and iT=-Ti in (9) to arrive at

i((∂I(t))/(∂t))+H^{†}(t)I(t)+I(t)H(t)=0,

which is the correct relation for the linear dynamical invariant I(t) when H(t) is non-Hermitian.

Eq. (10) does define a dynamical invariant when H(t) is a non-Hermitian Hamiltonian, in exactly the same way that Schrödinger equation applies to both Hermitian and non-Hermitian Hamiltonians. As a matter of fact, Eq. (10) is derived directly from the Schrödinger equation, as well as Eq. (9) for the symmetry operator (or an anti-Hermitian invariant for a non-Hermitian Hamiltonian). Other works in the literature share this claim:

[1] B. F. Ramos, I. A. Pedrosa, and A. Lopes de Lima, Eur. Phys. J. Plus 133, 449 (2018).
[2] W. Koussa, N. Mana, O. K. Djeghiour and M. Maamache, J. Math. Phys. 59, 072103 (2018).
[3] M. Maamache, O. Kaltoum Djeghiour, N. Mana, et al. Eur. Phys. J. Plus 132, 383 (2017).

Moreover, we note that the relation considered by the referee, H(-t)T=TH^{†}(t), when rewritten in the form

H^{†}(t)T-TH(-t)=i((∂T)/(∂t))=0,

becomes exactly the Eq. (19) of the new version of our manuscript:

H^{†}(t)Ξ(t)-Ξ(t)H(-t)=iΞ(t). <label>tdphr</label>

Therefore, by assuming the referee's relation, H(-t)T=TH^{†}(t), to be true, it is easy to see that we are assuming that the non-Hermitian H(t) is T-anti-pseudo-Hermitian. Consequently, the relation considered by the referee, i∂_{t}I(t)+H^{†}(t)I(t)+I(t)H(t)=0, is extremely particular compared to our Eq. (10) for a linear Invariant. Our liner invariant in Eq. (10) is general and valid for any non-Hermitian Hamiltonian.

8) The eight point raised by the referee is: Why do the authors decompose I(t) as the product of a time-dependent unitary or antiunitaryoperator Λ(t) and a linear operator U(t)? They could simply identify Λ(t) with the identity operator and T for cases where I(t) is linear and antilinear, respectively.

We have defined the symmetry operator as the product I(t)=Λ(t)U(t), with Λ(t) being either a unitary or non-unitary operator and U(t) an antilinear operator in accordance with the developments in Ref. [Phys. Rev. A 98, 032102 (2018)], where a method for the construction of nonlinear Lewis-Riesenfeld TD invariants is presented. As explained in the manuscript, by rewriting the Hamiltonian as H(t)=H₀(t)+V(t), with H₀(t) being either a diagonal or nondiagonal operator with known eigenstates, this decomposition, I(t)=Λ(t)U(t), is perfectly natural since U(t)=R(t)T accounts for H₀(t) through the operator R(t)=e^{iφ(t)H₀(t)}, while Λ(t) accounts for V(t). In fact, this decomposition proved to be quite efficient in the treatment of the Hamiltonians we approached, and also for the treatment of the Hermitian Hamiltonians considered in the cited reference.

9) The ninth point raised by the referee is: I find the analysis of Section V to be based on various ad hoc choices and formal manipulations. The basic physical motivation for the calculations reported in this section is missing. Do the results of these calculations lead to a concrete physical prediction about the behavior of the model the authors consider?

First of all, we must say that we do not know which section V the referee is referring to, from Part I or Part II of the previous version? Let us assume that the referee is referring to Part II of the previous version. Then, in the new merged version of the manuscript we make more clear our motivation for deriving an equation relating symmetry and metric, by changing the beginning of the last paragraph of Section IX to:

"Therefore, after computing the TD symmetry operator through Eq. (9), we automatically derive the Dyson map from the symmetry-metric Eq. (103), and consequently the metric operator or the ρ-pseudo-Hermitian operator. Then, from Eq. (110), we derive the Ξ-anti-pseudo-Hermitian operators."

Therefore, after computing the symmetry operator, all the other operators thus follow automatically, and this is a central result of our manuscripts. As we have explained in the manuscript "If the symmetry operator stems from the group algebra of the Hamiltonian, the ansatz for the Dyson's map must also be guided by this group algebra, as discussed in Ref. [11] where a strategy for constructing the Dyson map is presented. It is then reasonable to assume that there must be a close connection between both of these operators: the symmetry and the Dyson map (or equivalently the metric operator)."

We finally observe that the treatments of both cases, linear and parametric amplifications, were important to shed light on the connection between the symmetry operators associated with these Hamiltonians with their respective group algebra.
* * *
As a final observation, we would like to say that the manuscript brings new ingredients to the field of pseudo-hermiticity of TD Hamiltonians and metric operators. Our equation for the symmetry operator and its relation with the dynamic invariant is new in the literature, as well as our proposal for a relation between symmetry and metric. There are no formal errors in our manuscript. So, let's allow the community at large to judge our work.
* * *
Best Regards,
L. F. A. da Silva, R. A. Dorado, and M. H. Y. Moussa.

---

## Round 2 · List of Changes

A point-by-point list of changes is given above in the answers to the referee.

---

## Editorial Decision

published